# A continuum of zinc finger transcription factor retention on native chromatin underlies dynamic genome organization

Siling Hu [ID][1,2,3,4], Yangying Liu[1,2,3,4], Qifan Zhang [ID][1,2,3], Juan Bai[1,2,3] & Chenhuan Xu [ID][1,2,3✉]

## Abstract

**Transcription factor (TF) residence on chromatin translates into quantitative transcriptional or structural outcomes on genome. Commonly used formaldehyde crosslinking fixes TF-DNA interactions cumulatively and compromises the measured occupancy level. Here we mapped the occupancy level of global or individual zinc finger TFs like CTCF and MAZ, in the form of highly resolved footprints, on native chromatin. By incorporating reinforcing perturbation conditions, we established S-score, a quantitative metric to proxy the continuum of CTCF or MAZ retention across different motifs on native chromatin. The native chromatin-retained CTCF sites harbor sequence features within CTCF motifs better explained by S-score than the metrics obtained from other crosslinking or native assays. CTCF retention on native chromatin correlates with local SUMOylation level, and anti-correlates with transcriptional activity. The S-score successfully delineates the otherwise-masked differential stability of chromatin structures mediated by CTCF, or by MAZ independent of CTCF. Overall, our study established a paradigm continuum of TF retention across binding sites on native chromatin, explaining the dynamic genome organization.**

**Keywords** Native Chromatin; Chromatin Structure; CTCF; Transcription Factor; ChIP
**Subject Category** Chromatin, Transcription & Genomics

## Introduction

Transcription factors (TFs) play pivotal roles in gene expression regulation (Chen and Rajewsky, 2007; Spitz and Furlong, 2012; Suter, 2020). By occupying a set of cognate recognition DNA sequences in the genome, often summarized as a consensus motif sequence (Castro-Mondragon et al, 2022; Schneider and Stephens, 1990; Weirauch et al, 2014), TFs activate or repress gene transcription by respectively recruiting or repulsing the basic transcriptional machinery and the synergistically activating

complexes (Hanna-Rose and Hansen, 1996; Lambert et al, 2018; Reiter et al, 2017). Additionally, architectural type of TFs, such as CTCF, orchestrates the high-order chromatin structures by interacting with other TFs binding in distance, bringing the underlying *cis* regulatory elements in close spatial proximity, and thereby regulates gene expression in a broad scale (Di Giammartino et al, 2020; Jerković et al, 2020; Ortabozkoyun et al, 2022; Soochit et al, 2021; Tang et al, 2015; Zhang et al, 2016).

Different TFs have distinct residence time on chromatin, and even the same TF can have varying residence time across different genomic locations (Azpeitia and Wagner, 2020; Hansen et al, 2017; Lickwar et al, 2012), suggesting that the residence time is determined by principles of both the TF per se, and the local sequence feature and chromatin environment. The kinetics of TF residence on chromatin can translate into quantitative downstream transcriptional or structural outcomes (Lickwar et al, 2012; Soochit et al, 2021). Thus, knowing the relative residence time of a TF across different binding sites is critical for understanding its differential effects on genome organization and gene expression ultimately. Formaldehyde crosslinking can fix the protein-DNA interactions in the nucleus, and is widely used as a very first step in measuring the relative TF occupancy level on chromatin to proxy the TF residence time, through Chromatin Immunoprecipitation (ChIP) assays (Hoffman et al, 2015; Orlando et al, 1997; Park, 2009). In most ChIP practices, instead of presenting a snapshot image of protein-DNA interactions, the crosslinking step usually lasts for about 10 min, during which short-lived TF occupancy events may have a chance to be cumulatively fixed and finally appear in a ChIP dataset to be "highly occupied", compromising the measured spectrum of the TF's residence time across short-lived and long-lived binding sites (Poorey et al, 2013; Zaidi et al, 2017).

A few methods have been developed to measure the relative occupancy level of TFs or histones under native (without crosslinking) conditions (Brind'Amour et al, 2015; David et al, 2017; Grzybowski et al, 2019; Huang et al, 2020; Kaya-Okur et al, 2019; Skene and Henikoff, 2017). Most ChIP-based methods were developed for measuring occupancy of histones, which are presumably easier to retain on native chromatin than TFs (Brind'Amour et al, 2015; Grzybowski et al, 2019; Huang et al, 2020; Kaya-Okur et al, 2019). Other methods like CUT&RUN and

[1]CAS Key Laboratory of Genome Sciences and Information, Beijing Institute of Genomics, Chinese Academy of Sciences, Beijing, China. [2]China National Center for Bioinformatics, Beijing, China. [3]University of Chinese Academy of Sciences, Beijing, China. [4]These authors contributed equally: Siling Hu, Yangying Liu. ✉E-mail: xuchh@big.ac.cn

CUT&Tag circumvent the need for immunoprecipitation by in situ fragmenting the native chromatin where the protein of interest is present (Kaya-Okur et al, 2019; Meers et al, 2019; Skene and Henikoff, 2017). The above methods provide limited resolution of binding sites, and do not offer a tunable interface for explicitly delineating the dynamic and quantitative nature of TF-DNA interactions. Here we present two optimized methods for mapping the occupancy level of a global set of TFs or certain TF of interest on native chromatin, respectively. With the binding sites highly resolved in the form of very short footprints protected by TF binding, we further incorporated a series of reinforcing perturbing forces to uncover a continuum of retention of the same TF across different binding sites, which better explains the local sequence feature and the dynamic genome organization than the occupancy level obtained from crosslinking ChIP assays.

## Results

### A global set of high-resolution TF footprints on native chromatin

To comprehensively profile the *cis* regulatory elements and *trans* TFs across the entire human regulatory genome under native conditions, we fine-tuned the Micrococcal Nuclease digestion to relatively low degree, and performed library preparation and high-throughput sequencing (loMNase-seq) to obtain a genome-wide set of footprints protected by TF binding on native chromatin in chronic myelogenous leukemia cell line K562 (Henikoff et al, 2011; Iwafuchi-Doi et al, 2016) (Fig. EV1A). Although loMNase-seq largely recapitulates the regulatory chromatin landscape identified by commonly used DNase-seq or ATAC-seq (Boyle et al, 2008; Boyle et al, 2011; Buenrostro et al, 2013) (Fig. 1A), it still offers a few advantages in identifying TF footprints over the other two methods. Firstly, loMNase-seq generates uniformly shorter fragments, mostly ranging from 30 to 40 bp (Fig. 1B), exhibiting a more complete digestion than the other two methods. Secondly, loMNase-seq results have a higher fraction of fragments covering TF footprints (Fig. 1C), indicating a higher efficiency for mapping footprints than the other two methods. Lastly, loMNase-seq maps shorter fragments to TF footprints (Figs. 1D and EV1B), making fragment ends more approximate to the genuine protected regions, and thus achieving a higher-resolution definition of footprints than DNase-seq and ATAC-seq. In order to identify the TFs residing upon the footprints on the native chromatin, we performed de novo motif analysis using loMNase-seq peak regions. The top three highly represented motifs relate to the binding of ZNF143, MAZ, and CTCF (Figs. 1E and EV1C), all of which are zinc finger TFs well known for their roles in chromatin organization (Bailey et al, 2015; Chen et al, 2019; Ortabozkoyun et al, 2022; Tang et al, 2015; Xiao et al, 2021; Zhou et al, 2021). Additionally, loMNase-seq peaks overlap with many other TF peaks from ENCODE (Dunham et al, 2012) (Fig. EV1D,E). To investigate if binding of the three TFs left footprints in our loMNase-seq data, we utilized V-plot, a form of visual representation to uncover the existence of TF footprints, evidenced by the enzyme digestion boundaries (V-shaped vacancy of signal) and footprint-containing fragments (enrichment of inside-V signal) (Henikoff et al, 2011). As expected, there are typical "V" patterns on ZNF143, MAZ, and CTCF motifs, and on loMNase-seq peaks (Fig. 1C,F), strongly indicating that our loMNase-seq data embodies a rich repertoire of high-resolution TF footprints on native chromatin.

### Measuring the occupancy level of single TFs on native chromatin

To profile the occupancy of a TF of interest on native chromatin, we next streamlined low degree MNase digestion with chromatin immunoprecipitation, and developed a native ChIP (N-ChIP) strategy (Fig. EV2A). Due to the significant representation of CTCF, MAZ, and ZNF143 footprints in loMNase-seq data (Fig. 1E), we performed N-ChIP with antibodies targeting each of the three TFs. The N-ChIP results of the three TFs exhibit distinct relationship with results from formaldehyde-crosslinked ChIP (X-ChIP) (Fig. EV2B). The vast majority of CTCF N-ChIP peaks are included in the peak set discovered by CTCF X-ChIP (Fig. 2A,B). This subsetting effect suggests that our N-ChIP conditions exert strong selection on different CTCF binding sites. On the contrary, only 72% of MAZ, and 60% of ZNF143 N-ChIP peaks overlap with their respective X-ChIP peaks (Fig. 2A,B), suggesting that the N-ChIP conditions can have different selection effect on different types of TF-DNA complexes. The N-ChIP-specific peaks also suggest the potential advantage of native conditions in identifying epitope-masked TF binding sites due to formaldehyde crosslinking in X-ChIP (O'Neill and Turner, 2003). Next, we benchmarked the performance of our CTCF N-ChIP with other published CTCF antibody-based assays under native conditions in K562 cells (Kaya-Okur et al, 2019; Meers et al, 2019; Skene and Henikoff, 2017). Compared with CUT&RUN, CUT&Tag, and another published N-ChIP data, our CTCF N-ChIP data exhibit the highest efficiency of finding peaks under the same sequencing depth (Fig. 2C), and the highest ratio (93.8%) of CTCF motif-containing peaks (Figs. 2C and EV2C,D), suggesting that our N-ChIP conditions specifically capture chromatin complexes with CTCF binding directly to DNA. Furthermore, the fine-tuned MNase digestion enables our N-ChIP to more clearly uncover the digestion boundaries, and thus the genuine footprints protected by CTCF binding, when compared with other assays including ChIP-exo (Rhee and Pugh, 2011; Rossi et al, 2018) (Figs. 2D,E and EV2E).

### CTCF exerts a continuum of retention across different motif context on native chromatin

The facts that the most of CTCF N-ChIP peaks contain a CTCF motif, and are a subset of the X-ChIP peaks suggest that the native conditions select some CTCF-DNA complexes over the others, further raising an interesting scenario that CTCF proteins may have a continuum of retention across different sites on native chromatin, which could be experimentally delineated by a series of reinforcing perturbing forces. The salt, such as sodium chloride (NaCl), disrupts the electrostatic interactions between protein and DNA (Kasinathan et al, 2014; Yu et al, 2020). Thus, we performed parallel CTCF N-ChIP experiments under 75, 150, or 225 mM NaCl concentration during all the incubation and washing steps (Figs. 3A and EV3A,B). As we expected, the increasing salt concentration exert strong selection on different CTCF-DNA complexes. The peaks of each higher salt group are nested under the peaks of each lower salt group (Fig. 3B,C). The 225 mM group exhibits a highest 95.6% ratio of CTCF motif-containing peaks (Fig. EV3C). This consecutive "stripping" effect on CTCF proteins under salt elevation was also confirmed by western analysis of

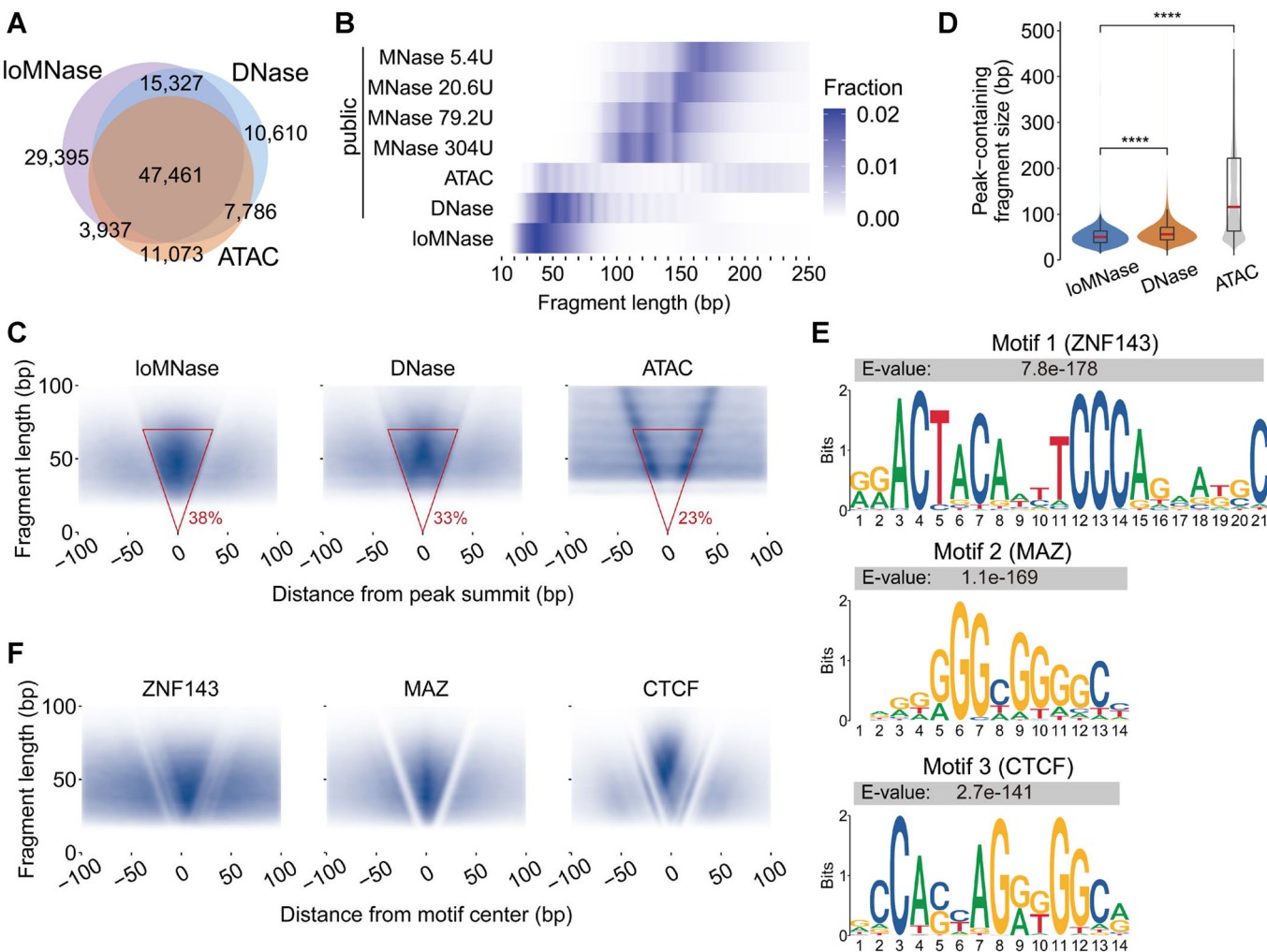

**Figure 1. loMNase-seq maps high-resolution TF footprints on native chromatin.**

(A) Venn diagram showing the overlap of peaks from loMNase-seq, ATAC-seq, or DNase-seq datasets. (B) Fragment length distribution of MNase-seq, ATAC-seq, DNase-seq, or loMNase-seq datasets. (C) V-plots of K562 loMNase-seq, DNase-seq, or ATAC-seq fragments at their respective peak summits. The numbers in red represent the percentage of fragments within the red triangle area out of the total fragments shown. (D) Violin plot and boxplot comparing the size difference of fragments containing peak summits in loMNase-seq, DNase-seq, or ATAC-seq datasets (loMNase-seq, $n = 682,305$; DNase-seq, $n = 6,520,460$; ATAC-seq, $n = 4,015,295$). For boxplot, the red central band represents the median. The lower and upper hinges represent the first and third quartiles, respectively. The whiskers represent the 1.5× interquartile range. ****$P < 0.0001$; two-sided Wilcoxon rank sum test. (E) Motif logos of the top three significant motifs identified by MEME-ChIP. (F) V-plots of loMNase-seq fragments at CTCF, MAZ, or ZNF143 motifs.

extracted chromatin fractions (Fig. EV3D). We next asked what sequence feature the remaining non-motif CTCF peaks have. Unexpectedly, we identified a variant form of CTCF motif (V-motif), with two adjacent bases in the canonical motif (C-motif) seemingly merged into one base (Fig. 3D). The set of V-motifs are occupied by CTCF in multiple cell lines (Fig. EV3E), and exhibit higher DNA bendability than C-motifs (Fig. EV3F) (Li et al, 2022), suggesting that they may be involved in genome organization across different cell types.

The canonical CTCF motif has been shown to occasionally have one upstream (U-motif) or downstream (D-motif) auxiliary motifs, which may enhance the occupancy by CTCF (Nakahashi et al, 2013; Rhee and Pugh, 2011; Schmidt et al, 2012; Soochit et al, 2021). In order to identify the sequence features of CTCF motifs associated with different extent of salt tolerance, we defined three groups of

CTCF motifs (N1, N2, N3) to reflect the subsequent loss and the retaining of occupancy by CTCF, when the salt concentration was incrementally increased (Fig. 3B). N1, the most refractory group, exhibits the highest frequency of possessing the U-motif, which is typically 5–6 bp upstream of the C-motif (Fig. EV3G), and is included in the footprints protected from MNase digestion (Fig. EV3H), while X-ChIP-specific motifs exhibit the lowest frequency (Fig. 3E,F). Additionally, the frequency of D-motif also weakly correlates with the extent of refractory of CTCF motifs to salt selection (Fig. 3E,F). The zinc finger domains of CTCF are proposed to exhibit a plasticity when engaged to different DNA sequence context (Nakahashi et al, 2013; Rhee and Pugh, 2011; Soochit et al, 2021), suggesting that CTCF proteins may have a spectrum of affinity at different sites, which can translate into a continuum of retention across different sites. To examine if this

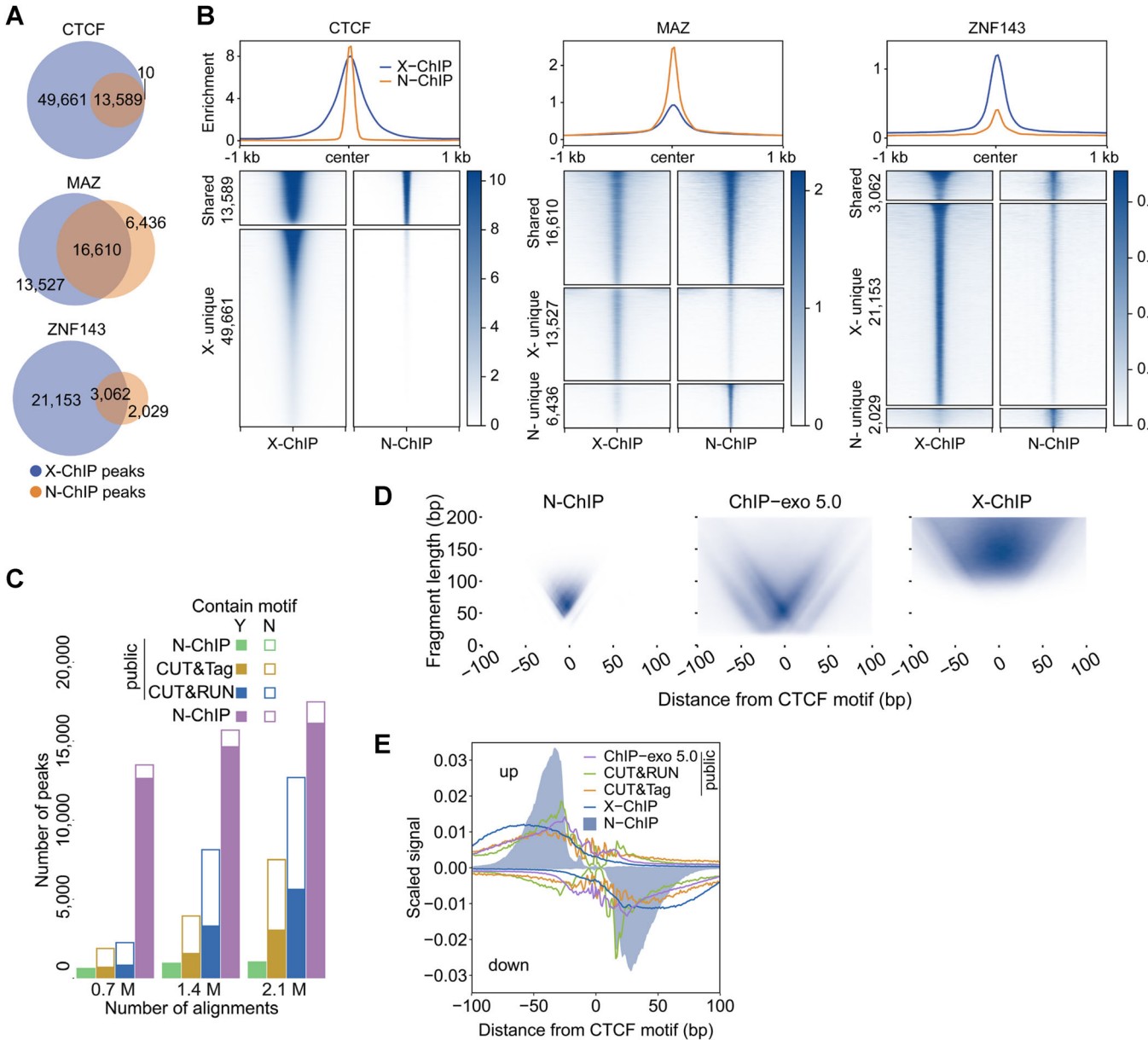

**Figure 2.  Occupancy of single TFs on native chromatin.**

(A) Venn diagrams showing overlap between X-ChIP and N-ChIP peaks for CTCF (150 mM NaCl), MAZ or ZNF143. (B) Average profiles and heatmaps showing normalized X-ChIP and N-ChIP signals centered around respective CTCF, MAZ, or ZNF143 peaks defined in (A). (C) Barplot showing the number of peaks identified with 0.7, 1.4 or 2.1 million alignments in different datasets. The fractions of peaks containing CTCF motifs (Y) or not (N) are indicated. (D) V-plots of CTCF N-ChIP (150 mM NaCl), ChIP-exo 5.0 (published) and X-ChIP fragments at CTCF motifs. (E) The scaled pileup of fragment 5′ and 3′ ends around CTCF motifs from ChIP-exo, CUT&RUN, CUT&Tag, X-ChIP or N-ChIP datasets. All alignments were oriented in the same direction relative to their nearest CTCF motif.

possibility underlies CTCF motif's differential refractory to salt selection, we built a quantitative "Salt Tolerance Score" (S-score) for each CTCF motif using our CTCF N-ChIP data under three salt concentrations (Fig. EV3I and see "Methods"). As expected, ranking of CTCF motifs by the S-score reveals an association with the presence of auxiliary motifs (Figs. 3F,G and EV3J,K), and performs better than ranking by X-ChIP, CUT&RUN, or CUT&Tag signal, in correlating with the consensus U-motif (Fig. 3H), suggesting that CTCF proteins possess a continuum of retention across different motif sequence context, which can be

better delineated by the salt elevation and the association with the auxiliary motifs on native than on crosslinked chromatin.

## CTCF retention on native chromatin associates with transcriptional activity and post-translational modifications

We next asked what chromatin features are concordant with the continuum of CTCF retention across different motif context. The above defined N1, N2, and N3 groups representing the series of

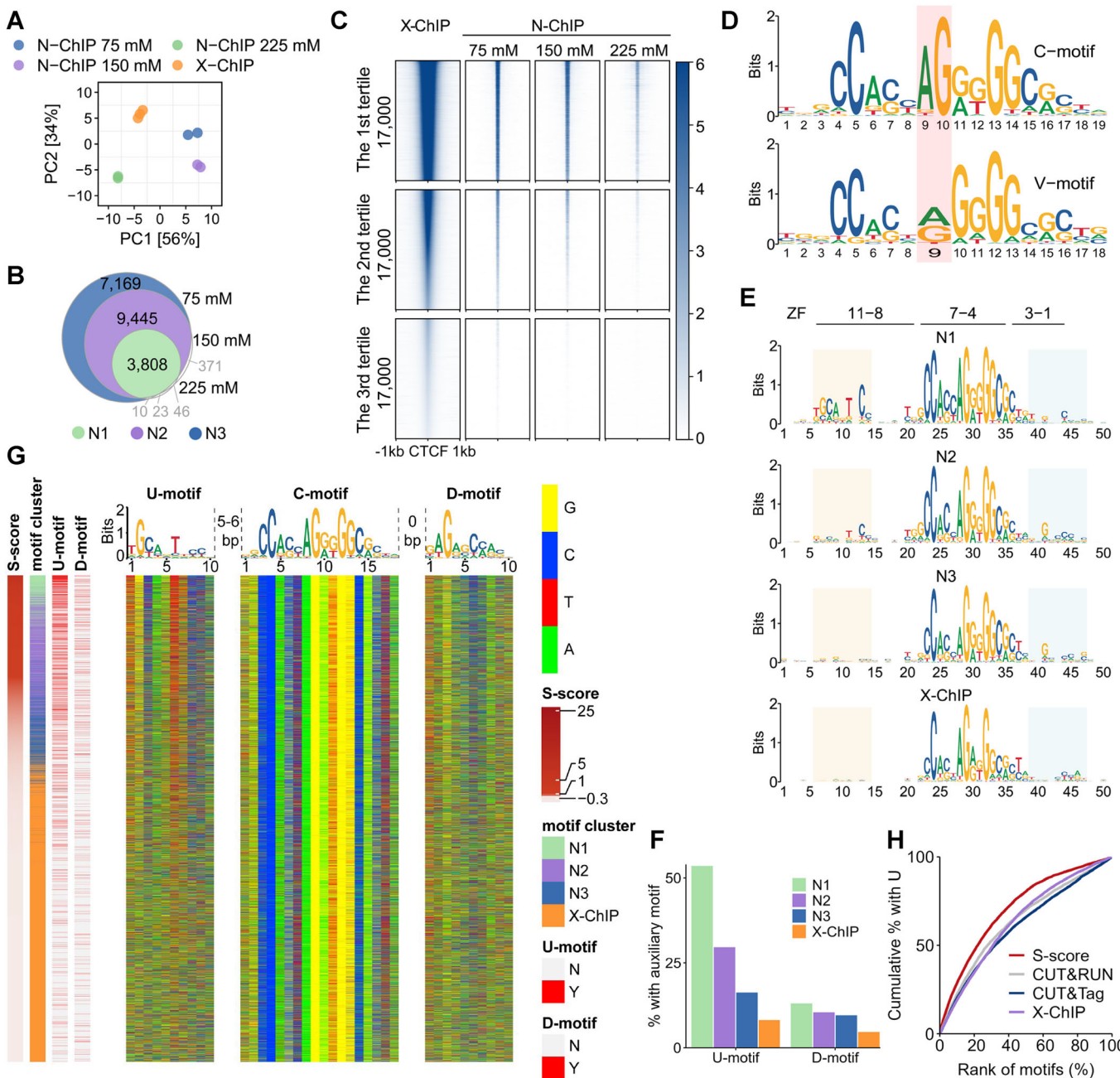

**Figure 3. A continuum of CTCF retention across genomic sites on native chromatin.**

(A) PCA of CTCF X-ChIP and N-ChIP datasets. Each dot represents an individual replicate. (B) Venn diagram showing the overlap between 75, 150, and 225 mM CTCF N-ChIP peaks. The N1, N2, and N3 sites are classified as indicated. (C) Heatmaps showing normalized CTCF X-ChIP and N-ChIP signals centered around the three tertiles of CTCF motifs with high, medium, or low X-ChIP signal that are within CTCF peaks identified in X-ChIP and N-ChIP datasets. (D) Motif logo representation of CTCF canonical motif (C-motif) and variant motif (V-motif) contained in native CTCF sites. (E) Motif logo representation of C-motifs in the top 100 N1, N2, N3, or X-ChIP specific sites. Yellow and blue shadings are for upstream and downstream motif regions, respectively. The zinc finger (ZF) domains binding to these regions are indicated. (F) Percentage of C motif-containing N1, N2, N3 and X-ChIP sites that harbor a U-motif or a D-motif. (G) The base composition of CTCF motif regions within the N1, N2, N3, and X-ChIP peaks. Rows are ordered by the S-score of the C-motif. The left four tracks represent the S-score, motif clusters, and the presence (Y) or absence (N) of a U- or D-motif respectively. Motif logos and location of the U-, C-, and D-motifs are shown at the top. Yellow, blue, red and green represent G, C, T, and A bases, respectively. (H) Percentage plot showing accumulation of ranks of CTCF motifs with a U-motif. The motifs are ranked in descending order based on S-score or normalized signals from CUT&RUN, CUT&Tag, or X-ChIP. X axis represents the rank of motifs divided by the total number of motifs.

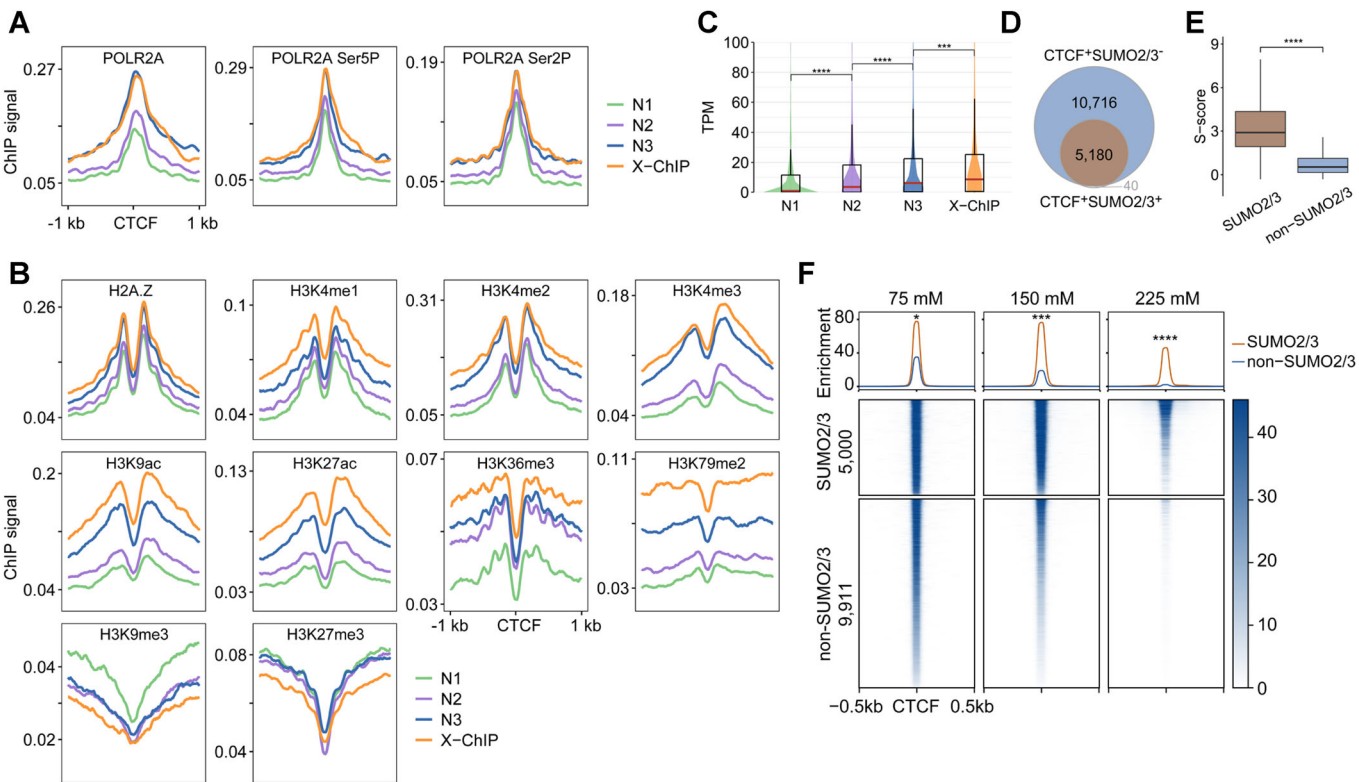

**Figure 4. CTCF retention on native chromatin associates with transcriptional activity and post-translational modifications.**

(A) Average profiles showing ChIP signals for different RNA polymerase II forms (pan-Pol II, Ser2 or Ser5 phosphorylated Pol II) centered around sampled CTCF motifs from N1, N2, N3 and X-ChIP groups. (B) Average profiles showing ChIP signals for different histone modifications centered around sampled CTCF motifs from N1, N2, N3 and X-ChIP groups. (C) Gene expression levels at sampled CTCF motifs from N1, N2, N3 and X-ChIP groups (N1, $n = 1568$; N2, $n = 1709$; N3, $n = 1832$; X-ChIP, $n = 2087$). ****$P < 0.0001$; two-sided Wilcoxon rank sum test. (D) Venn diagram showing the overlap between CTCF$^+$SUMO2/3$^+$ and CTCF$^+$SUMO2/3$^-$ peaks. SUMO2/3 (brown) and non-SUMO2/3 (blue) sites were defined based on the Venn diagram. (E) Boxplot showing the S-score of CTCF motifs within SUMO2/3 and non-SUMO2/3 sites. ****$P < 0.0001$; two-sided Wilcoxon rank sum test. (F) Average profiles and heatmaps representing normalized CTCF N-ChIP signals under 75, 150, or 225 mM NaCl concentration centered around CTCF motifs within SUMO2/3 and non-SUMO2/3 sites. *$P < 0.05$, ***$P < 0.001$, ****$P < 0.0001$; two-sided $t$ test. Data information: For boxplots in (C, E), the central band represents the median. The lower and upper hinges represent the first and third quartiles, respectively. The whiskers represent the 1.5× interquartile range.

CTCF motifs refractory to salt elevation, along with the X-ChIP-specific (X) motifs, were used to examine the co-localization with various ChIP signals from the GEO (Barrett et al, 2013) and ENCODE (Dunham et al, 2012) databases. After sampling to reach a consistent CTCF X-ChIP signal (Fig. EV4A), as expected, the four groups show a gradual decrease of S-score from N1 to X (Fig. EV4B), reflecting their decreasing retention on native chromatin. The occupancy levels of all RNA polymerase II forms (pan-Pol II, Ser2- or Ser5-phosphorylated Pol II) anti-correlate with the retention ability of CTCF motif groups (Fig. 4A). All the examined histone modifications associated with transcriptional initiation (H2A.Z, H3K4me2, H3K4me3) (Zhou et al, 2011), elongation (H3K36me3, H3K79me2) (Gates et al, 2017), or enhancers (H3K4me1, H3K9ac, H3K27ac) (Kundaje et al, 2015; Zhang et al, 2020), also anti-correlate with CTCF retention on native chromatin, while H3K9me3 and H3K27me3, the two marks for repressive chromatin (Kundaje et al, 2015), slightly correlate with CTCF retention (Fig. 4B). By restricting with higher retention ability, CTCF motifs tend to locate at regions with little transcriptional or enhancer activity (Figs. 4C and EV4C–E). The above results suggest that the processive transcription process may

increase the turnover rate of binding of the encountered CTCF proteins, and thus reduce their retention time on chromatin (Heinz et al, 2018; Marina-Zárate et al, 2023; Soochit et al, 2021).

In an effort to identify associated non-histone proteins, we found that chromatin SUMOylation level also correlates with CTCF retention (Fig. EV4F). CTCF proteins are known to harbor SUMOylated residues which may alter their function (Kitchen and Schoenherr, 2010; MacPherson et al, 2009). In order to confirm that the observed correlation occurs on the same CTCF chromatin complexes, we designed a sequential N-ChIP strategy by sequential pull-downs using CTCF and SUMO2/3 antibodies (Fig. EV4G,H). By preparing and sequencing libraries from both fractions after the 2nd pull-down, we found that most of the CTCF$^+$SUMO2/3$^+$ peaks are nested under the set of CTCF$^+$SUMO2/3$^-$ peaks (Fig. 4D). The CTCF motifs from CTCF$^+$SUMO2/3$^+$ peaks have significantly higher S-score than the motifs from CTCF$^+$SUMO2/3$^-$-specific peaks (Fig. 4E), and mostly retained CTCF occupancy on native chromatin under the most stringent 225 mM salt concentration (Fig. 4F). Notably, 98.6% of the SUMO2/3 peaks contain at least one CTCF motif (Fig. EV4I), suggesting that our N-ChIP conditions keep a rich content of direct and very little indirect

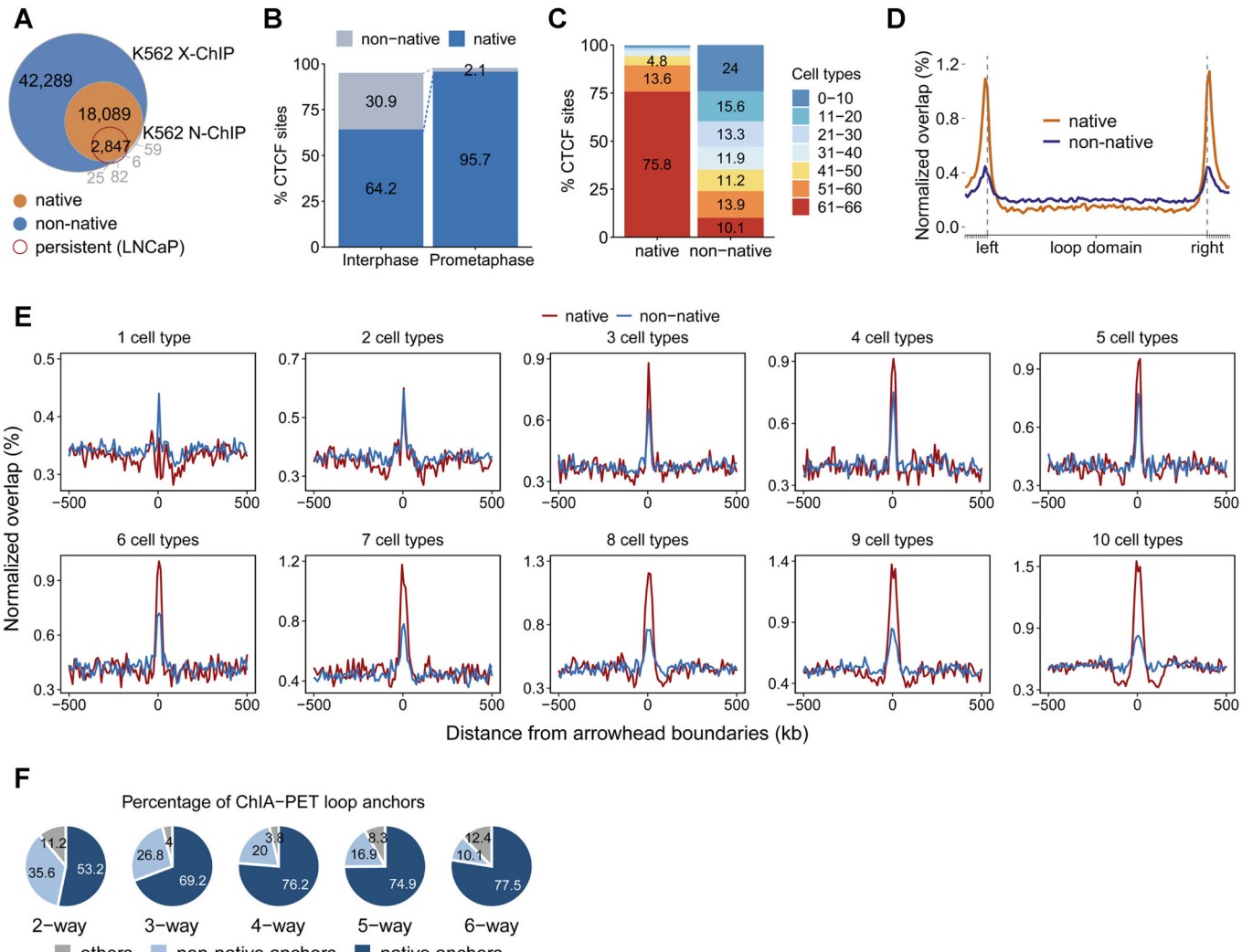

**Figure 5. Native chromatin-retained CTCF sites exhibit functional conservation.**

(A) Venn diagram showing the overlap between CTCF X-ChIP, N-ChIP peaks, and published LNCaP persistent sites. (B) Percentage of native and non-native sites that intersect with published U2OS CTCF peaks in interphase and prometaphase. (C) Bar graph showing proportions of native or non-native sites exhibiting constitutive occupancy by CTCF in multiple cell lines. (D) Profile of native sites and non-native sites at K562 loop domains from published Hi-C dataset. Y axis depicts the percentage of overlapping CTCF binding sites per 1000 loop domains. (E) Domain boundaries were grouped based on whether each boundary was present in 1–10 different cell lines. Positional enrichment of native or non-native CTCF sites was plotted for each group. Y axis depicts the percentage of overlapping CTCF binding sites per 1000 domain boundaries. (F) Percentage of native and non-native sites in anchors of pairwise (2-way) or multi-way contacts (3-, 4-, 5-, and 6-way).

TF-DNA interactions on native chromatin. Thus, the SUMOylation we observed very likely occurs on CTCF proteins. We used a small compound ML-792 to inhibit SUMOylation (Schneeweis et al, 2021), and observed no obvious changes in CTCF N-ChIP signals (Fig. EV4J), suggesting that SUMOylation may be a consequence instead of a cause for CTCF retention on native chromatin.

## Native chromatin-retained CTCF sites exhibit functional conservation

Next, we investigated if the native CTCF sites can also be retained on chromatin under perturbing forces other than salt. A recent report identified a subset of CTCF binding sites relatively persistent under RNAi reduction of CTCF protein level (Khoury et al, 2020). When compared with native and non-native CTCF binding sites

defined in this study, the RNAi persistent sites form a subset of our native sites (Fig. 5A; Appendix Fig. S1A,B). Likewise, 95.7% of a previously reported set of mitotically bookmarked CTCF binding sites overlap with our native sites (Kang et al, 2020) (Fig. 5B; Appendix Fig. S1C). These results suggest a functional relevance between TF's refractory to salt elevation, resistance to protein depletion, and retention on highly condensed chromatin. To examine if this functional relevance of CTCF binding sites extends into a functional conservation across cell types, we analyzed CTCF ChIP-seq peaks from 66 different cell types, and Hi-C/Micro-C datasets from 10 different cell types from ENCODE (Dunham et al, 2012), GEO (Barrett et al, 2013) and 4DN (Dekker et al, 2017) databases. 75.8% of the native sites are constitutively occupied in more than 60 cell types (Fig. 5C; Appendix Fig. S1D). Compared with the non-native sites, the native sites are more enriched at

chromatin loop anchors (Fig. 5D), and possess a higher fraction of inward-interacting CTCF motifs (Appendix Fig. S1E). Specifically, the native sites are more represented at domain boundaries when the boundaries become conserved across more cell types (Fig. 5E; Appendix Fig. S1F). Furthermore, the native sites tend to be over-represented in chromatin interaction complexes composed of multiple *cis* regulatory elements (Fig. 5F; Appendix Fig. S1G). The functional conservation across different cellular activities and cell types suggests that some intrinsic features of CTCF protein and local chromatin may help retain CTCF proteins at these binding sites.

## TFs display differential susceptibility to zinc depletion on native chromatin

CTCF, MAZ, and ZNF143, the top three TFs we identified on native chromatin through loMNase-seq, are all zinc finger TFs, suggesting that their retention on native chromatin may require the zinc finger domains which are stabilized by zinc ions (Bailey et al, 2015; Nakahashi et al, 2013; Song et al, 2001; Soochit et al, 2021; Xiao et al, 2021) (Fig. 1E,F). To investigate if the three TFs exhibit the same or differential dependence on zinc finger domains, we treated K562 cells with Tris(2-pyridylmethyl)amine (TPA), a zinc ion-specific chelating agent (Huang et al, 2013), to deplete zinc ions in the nucleus, and performed loMNase-seq to examine the global TF occupancy changes on native chromatin (Fig. 6A; Appendix Fig. S2A–C). CTCF appears to be more susceptible to zinc ion depletion than TFs like MAZ or ZNF143, accounting for 66.3% of lost loMNase-seq peaks under TPA treatment (Fig. 6A–C; Appendix Fig. S2D), while MAZ or ZNF143 motif-containing peaks are relatively stable (Fig. 6B,C). Using the S-score defined previously (Fig. 3G), we found that occupancy at CTCF motifs exhibits diminishing resistance to zinc ion depletion when the motif S-score decays (Fig. 6D; Appendix Fig. S2E). To confirm that the loss of occupancy at CTCF motifs in loMNase-seq data were indeed caused by loss of CTCF occupancy, we performed CTCF N-ChIP under TPA treatment (Appendix Fig. S2F,G). Compared with a 25.7% fraction of lost peaks in loMNase-seq data (Fig. 6B), CTCF N-ChIP data exhibits a larger 35.4% fraction of lost peaks under TPA treatment (Fig. 6E; Appendix Fig. S2H). Interestingly, the stable peaks embody higher fractions of CTCF motifs with auxiliary U- or D-motif than the lost peaks (Fig. 6F; Appendix Fig. S2I). Ranking by motif S-score better explains the stable peaks under TPA treatment than ranking by signals from X-ChIP, CUT&RUN, or CUT&Tag assays (Fig. 6G), pinpointing an accordance between salt elevation and zinc ion depletion in delineating CTCF retention on native chromatin, which is consistent with the functional conservation exhibited by the native chromatin-retained CTCF sites (Fig. 5A–F).

## The continuum of CTCF retention translates into concordant stability of chromatin structures under zinc depletion

The TPA treatment reduced the occupancy level of CTCF more than other TFs (Fig. 6A). To investigate if this perturbation effect relatively specific to CTCF can translate into an alteration to CTCF-mediated genome organization, we performed Micro-C, the newest generation of Hi-C technology, in K562 cells with or without TPA

treatment (Appendix Fig. S3A,B). As in the case of acute degradation of CTCF proteins (Nora et al, 2017), the global chromatin interaction distance and the compartmental segregation of genome do not change significantly (Appendix Fig. S3C–E). The contact domain boundaries exhibit a CTCF-specific effect under TPA treatment, with CTCF motif-containing boundaries showing greater reduction of insulation effect than boundaries without CTCF motif (Fig. 7A; Appendix Fig. S3F). Specifically, at boundaries where a pair of divergent CTCF motifs insulate two adjacent contact domains, the strength of CTCF motif S-score correlates with the refractory to the weakening of boundaries (Fig. 7B; Appendix Fig. S3G). To further delineate this correlation, we analyzed the interaction changes at the series of six CTCF motif groups showing continuous decrease of S-score and retention on chromatin (Fig. 6D). As expected, the interactions emanating from CTCF motifs across the six groups exhibit similar frequency, but become more severely reduced under zinc depletion when the motifs have less retention ability on chromatin (Fig. 7C,D). To gain a more quantitative assessment, we compared the performance of ranking by S-score or X-ChIP signal in explaining the stable boundaries. Interestingly, after CTCF motifs are divided into two groups by the S-score, only the higher half percentile of S-score performs better than the respective X-ChIP metric in correlating with stable boundaries (Fig. 7E). Likewise, CTCF-mediated chromatin loops become more susceptible to zinc depletion when the motifs exhibit less retention ability (Fig. 7F,G). The above results suggest that the continuum of CTCF retention delineated on native chromatin translates into a functional consequence of dynamic CTCF-mediated chromatin structures, which is usually masked by crosslinking-based Micro-C assays, and can be revealed by perturbation conditions like salt elevation or zinc depletion.

## The S-score delineates MAZ retention on native chromatin and stability of MAZ-mediated chromatin loops independent of CTCF

In order to showcase the expandability of our N-ChIP method and the analytical framework using S-score, we performed a similar series of MAZ N-ChIP experiments under 50, 100, or 150 mM NaCl concentration (Fig. EV5A). As in the case of CTCF N-ChIP, the MAZ peaks from each higher salt group are nested under the MAZ peaks from each lower salt group (Fig. EV5B,C). Based on this subsetting effect, we established a similar S-score for MAZ to proxy its retention on native chromatin (see "Methods"), and defined four groups of MAZ motifs with similar X-ChIP signals but gradual decrease in S-score from M1 to M4 (Fig. EV5D). While the conservation of MAZ core motif sequence does not have an association with the S-score, the MAZ motifs with higher S-score tend to be flanked by higher content of G (Fig. EV5E). In contrary to CTCF, the highly retained MAZ motifs associate with higher levels of transcription and transcription-related histone modifications (Fig. EV5F). In agreement with this, the highly retained MAZ motifs tend to have low co-occupancy level of CTCF and Cohesin (Fig. EV5G). The MAZ S-score does not appear to correlate with differential stabilities of MAZ-mediated chromatin loops under TPA treatment (Fig. EV5H), possibly due to compensation from other TFs, like CTCF. Indeed, after excluding Micro-C contacts close to CTCF motifs, the MAZ S-score successfully delineates a spectrum of stability of MAZ-mediated chromatin loops

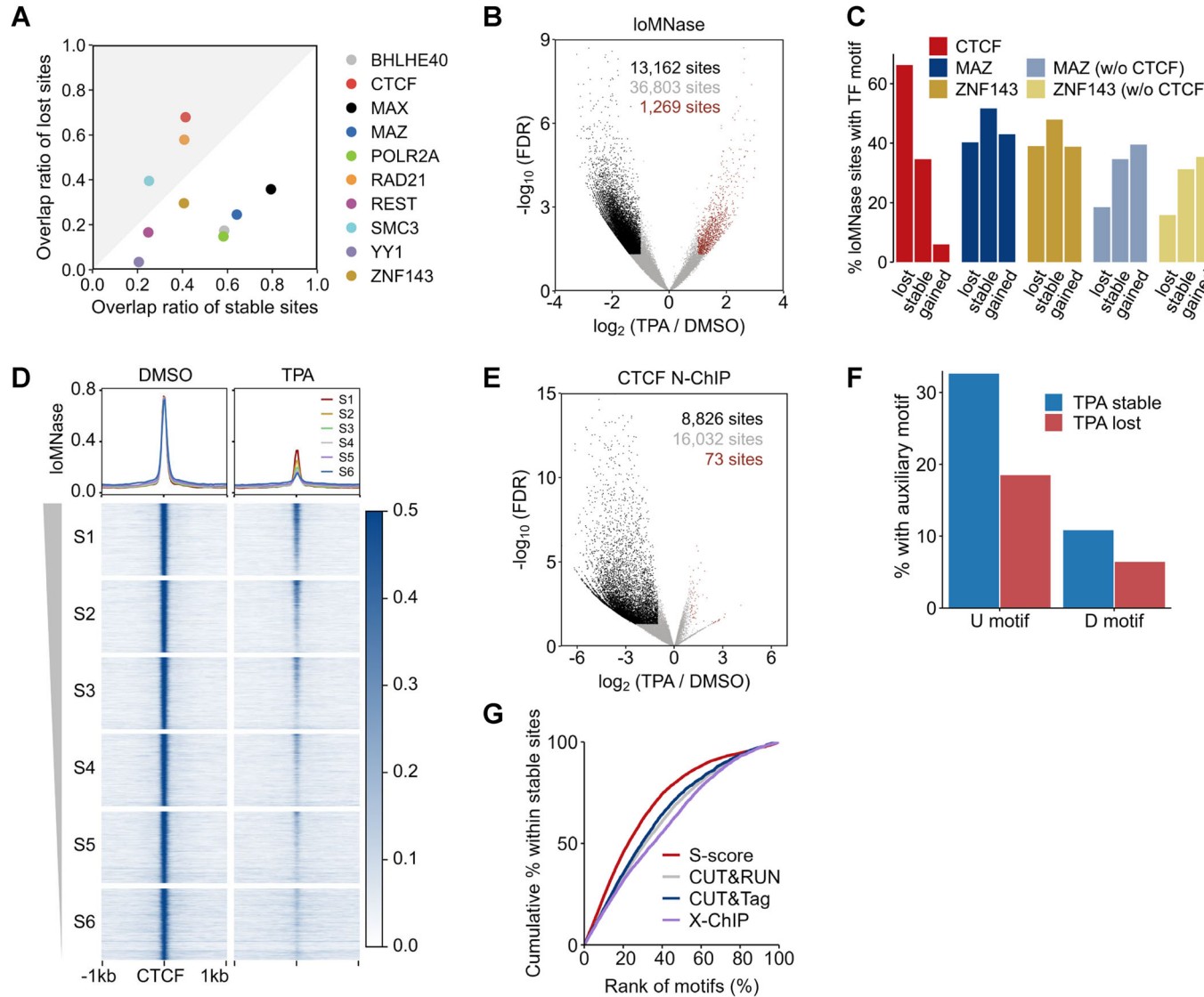

**Figure 6. TFs display differential susceptibility to zinc depletion on native chromatin.**

(A) Scatter plot showing overlap ratio of lost or stable loMNase-seq peaks with peaks of TFs shown. (B) Volcano plot showing the loMNase-seq differential peaks between DMSO and TPA conditions. Black, gray, and red indicate lost, stable, and gained sites after TPA treatment, respectively. (C) Percentage of lost, stable, or gained loMNase-seq peaks overlapping with CTCF, MAZ, or ZNF143 motifs. (D) Average profiles and heatmaps showing normalized loMNase-seq signals under DMSO or TPA conditions centered around six groups of CTCF motifs defined in Appendix Fig. S2E. (E) Volcano plot showing the CTCF N-ChIP differential peaks between DMSO and TPA conditions. Black, gray, and red indicate lost, stable, and gained sites after TPA treatment, respectively. (F) Proportion of CTCF motif-containing lost or stable sites harboring a U- or D-motif. (G) Percentage plot showing accumulation of ranks of CTCF motifs within stable sites. The motifs are ranked in descending order based on S-score or normalized signals from CUT&RUN, CUT&Tag, or X-ChIP. X axis represents the rank of motifs divided by the total number of motifs.

independent of CTCF (Fig. EV5H,I), with more highly retained MAZ motifs residing at anchors of less susceptible loops under TPA treatment. The above results suggest that different zinc finger TFs, like CTCF and MAZ, may act through division of labor, to regulate dynamic genome organization.

## Discussion

The in situ formaldehyde crosslinking of TF-DNA interactions in nucleus is a relatively inefficient reaction (Schmiedeberg et al,

2009), rendering ten-minute-long crosslinking step a necessity for most assays probing chromatin activities involving TF-DNA interactions, such as ChIP or Hi-C. Within 10 min, the diffusive nature of formaldehyde, a very small molecule, can introduce stochasticity in crosslinking different TF-DNA complexes in the nucleus. Furthermore, CTCF proteins have been determined to have a residence time of a few minutes on chromatin (Hansen et al, 2017), while the measured residence time for many other TFs are at sub-minute scale (Hsieh et al, 2022; Liu and Tjian, 2018; Morisaki et al, 2014; Zaret et al, 2016), an order of magnitude less than the most commonly used formaldehyde crosslinking duration.

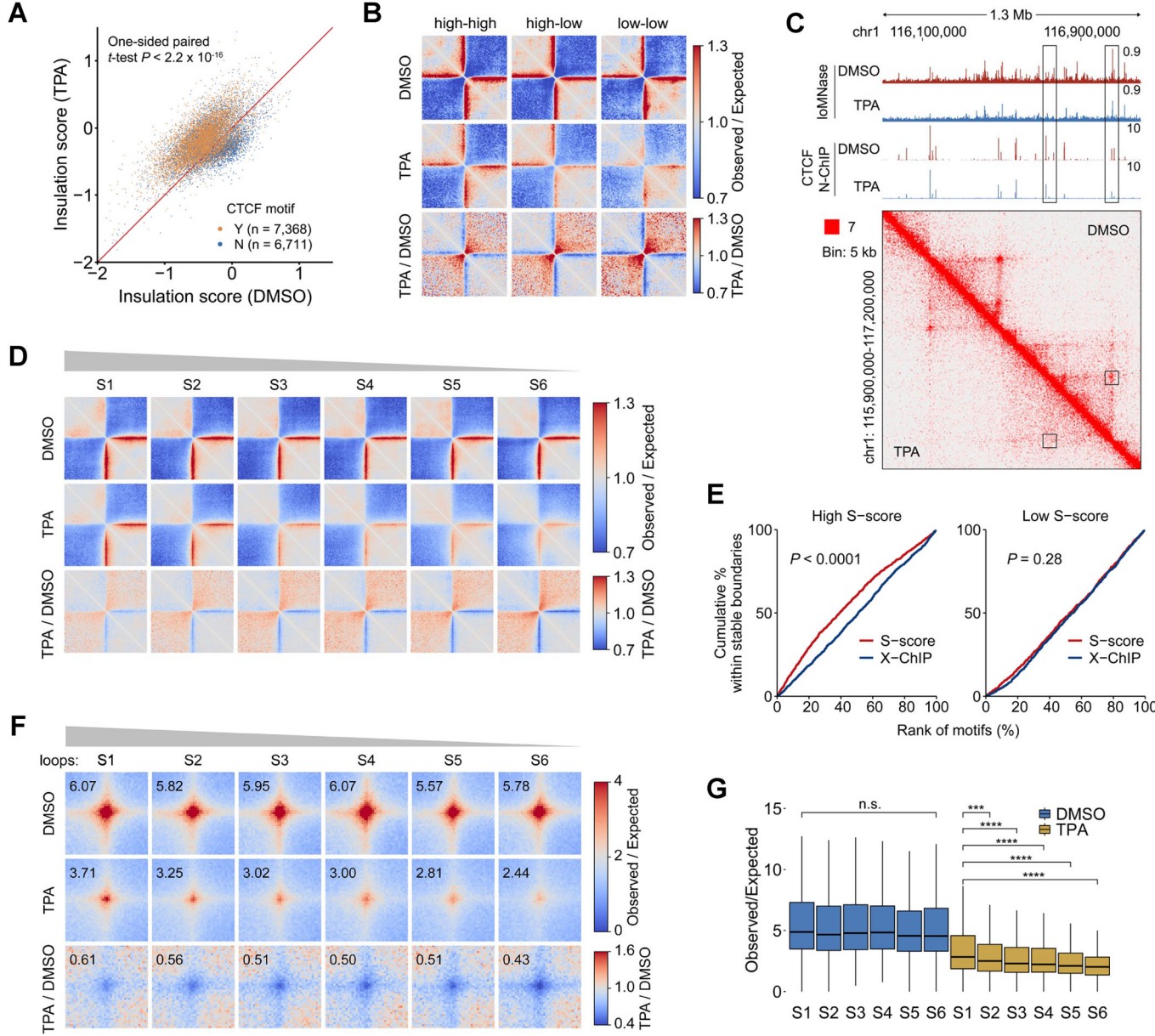

**Figure 7. The continuum of CTCF retention translates into concordant stability of chromatin structures under zinc depletion.**

(A) Scatter plot showing changes of insulation score of boundaries containing CTCF motifs (Y) or not (N) under TPA treatment. P value is calculated by a one-sided paired Student's t test. (B) Local pileup plot of long-range interactions using Micro-C data under DMSO or TPA conditions at subsets of boundaries containing divergent CTCF motifs grouped with different S-score combinations. Each plot shows a 1 Mb region centered on the boundaries. (C) A screenshot showing changes of loMNase-seq, CTCF N-ChIP and Micro-C signals between DMSO and TPA conditions. (D) Local pileup plot of long-range interactions using Micro-C data under DMSO or TPA conditions at the six groups of CTCF motifs classified in Appendix Fig. S2E. Each plot shows a 1 Mb region centered on different groups of CTCF motifs. (E) Percentage plots showing accumulation of ranks of CTCF motifs within stable boundaries. The two half percentiles of motifs with higher or lower S-score are shown respectively. The motifs are ranked in descending order based on S-score or normalized X-ChIP signals. X axis represents the rank of motifs divided by the total number of motifs. P value is calculated by two-sided Wilcoxon rank sum test. (F) APA plots showing the changes of chromatin loop strength under TPA treatment. The loops are grouped based on the presence of CTCF motifs from the six groups at their anchors, as defined in Appendix Fig. S2E. Bin size: 5 kb. (G) Boxplot of observed over expected scores of loops as described in (F). Numbers of loops (from left to right): $n = 844/812/748/743/781/849/844/812/748/743/781/849$. The central band represents the median. The lower and upper hinges represent the first and third quartiles, respectively. The whiskers represent the 1.5× interquartile range. n.s., not significant; ***$P < 0.001$, ****$P < 0.0001$; two-sided Wilcoxon rank sum test.

Therefore, a single formaldehyde crosslinking step can substantially fix and accumulate short-lived TF occupancy events, and compromise the measured spectrum of residence time across different TFs, and across different genomic sites for the same TF (Poorey et al, 2013).

CUT&RUN and CUT&Tag, performed under native conditions with simplified and streamlined procedures (Kaya-Okur et al, 2019; Meers et al, 2019; Skene and Henikoff, 2017), stand as good alternatives for X-ChIP, and particularly facilitate low-input and automation-based profiling assays (Skene et al, 2018). In this study,

we aimed to develop a standard version of N-ChIP strategy to probe into the spectrum of TF retention on native chromatin. We first optimized the MNase digestion condition and developed loMNase-seq, and obtained a genome-wide set of TF binding sites in the form of high-resolution footprints on native chromatin. Using this dataset as a preliminary screening, we identified CTCF, MAZ, and ZNF143 as the most highly represented TFs on native chromatin. By incorporating the antibody pull-down step, we succeeded in profiling occupancy level of the three individual TFs on native chromatin. Compared with a static snapshot of genome-wide TF-DNA interactions provided by a single ChIP experiment, a more quantitative and dynamic metric of TF-DNA interactions can be delineated by a series of ChIP experiments performed under reinforcing perturbation conditions (Henikoff et al, 2009). Under increasing salt concentration, CTCF or MAZ occupancy on native chromatin exhibit a strong subsetting effect, with smaller subset of motifs retain TF occupancy when the salt concentration becomes higher. Using the differential refractory to salt elevation, we further built a quantitative S-score for CTCF or MAZ motifs to proxy their continuum of retention across different motifs on native chromatin. The S-score performs better than the metrics obtained from X-ChIP, CUT&RUN, or CUT&Tag assays, in explaining the signature sequence feature of highly retained CTCF motifs, and the refractory to zinc depletion, another perturbing force, across different CTCF motifs.

The interactions between chromatin-residing TFs serve as the basis of high-order chromatin structures (Hansen et al, 2017; Kim and Shendure, 2019). The growing collection of datasets profiling chromatin structures across different cell or tissue types using Hi-C technologies were mostly acquired under similar crosslinking conditions as in ChIP assays (Lieberman-Aiden et al, 2009; Rao et al, 2014). Completely fixing a chromatin interaction complex may take more formaldehyde-mediated reactions than fixing a single TF-DNA occupancy event. Indeed, crosslinking step often lasts for almost an hour in the newly developed Micro-C assay (Hsieh et al, 2020; Krietenstein et al, 2020). Therefore, the measured interaction frequency across different TF binding sites may somehow been compromised by crosslinking as well (Gavrilov et al, 2015). Interestingly, the S-score built upon CTCF or MAZ occupancy on native chromatin successfully unmasks the gradient of refractory to zinc depletion across different motifs underlying chromatin structures profiled by crosslinking Micro-C assays, suggesting that the quantitative metric of interaction frequency can somehow be further delineated by perturbation experimental conditions, even under crosslinking conditions. Nevertheless, future development of new methods employing the same principle of ligating fragments in close spatial proximity as in Hi-C, but performed under native conditions, will ultimately help elucidate the bona fide quantitative metric of interaction frequency across genomic sites.

In summary, loMNase-seq and N-ChIP, the two optimized methods we present in this study, along with the experimental setup under perturbing forces, provide a paradigm framework for analyzing and delineating the continuum of TF retention on native chromatin and its outcomes on genome organization, and hold a promise for establishing the causal relationship between TF residence and local chromatin features, such as transcriptional activity or post-translational modifications.

# Methods

## Cell culture

K562 cells (CCL-243, ATCC) were cultured in RPMI 1640 medium (C3010-0500, Vivacell) supplied with 10% Fetal bovine serum (04-001-1A, Biological Industries) under 5% $CO_2$ at 37 °C. Tris(2-pyridylmethyl)amine (TPA) (Y24201, Alfa Aesar) stock was prepared by dissolving in DMSO (D2650, Sigma) to a final concentration of 10 mM. For TPA treatment, K562 cells were seeded and cultured overnight to 80% density, supplemented with TPA at a final concentration of 100 μM, or with DMSO as negative control, and treated for 1 h. For ML-792, K562 cells were seeded and cultured overnight to 80% density, supplemented with ML-792 (HY-108702, MedChemExpress) at a final concentration of 100 or 400 nM, or with DMSO as negative control, and cultured for 72 h. The cell line was authenticated and tested for mycoplasma contamination regularly.

## Chromatin extraction

About $4–5 \times 10^6$ cells were collected by centrifugation at 4 °C for 3 min at $500 \times g$, washed with 1× PBS/1 mM EDTA. After centrifugation, the plasma membranes were lysed by resuspension in ice-cold NP-40 lysis buffer (10 mM Tris-HCl [pH 7.5], 0.01% NP-40, 75 mM NaCl) and incubated on ice for 3 min. The lysate was then layered on top of 2.5 volumes of a chilled sucrose cushion (24% sucrose in lysis buffer, no NP-40) and centrifuged at 4 °C for 3 min at $800 \times g$. The nuclei pellet was gently rinsed with ice-cold 1× PBS/1 mM EDTA, and then resuspended in a pre-chilled glycerol buffer (20 mM Tris-HCl [pH 7.9], 75 mM NaCl, 0.5 mM EDTA, 0.85 mM DTT, 0.125 mM PMSF, 50% glycerol) by gentle flicking of the tube. An equal volume of cold nuclei lysis buffer (10 mM HEPES [pH 7.6], 1 mM DTT, 7.5 mM $MgCl_2$, 0.2 mM EDTA, 75/225/375 mM NaCl, 1 M UREA, 0.1% NP-40) was added, and the tubes were gently rotated for 1 h at 4 °C, and centrifuged at 4 °C for 3 min at $800 \times g$. The supernatant was collected. Finally, the nuclei pellet was resuspended in a pre-chilled glycerol buffer and an equal volume of cold nuclei lysis buffer (1% NP-40 instead) and rotated for 15 min at 4 °C. Then the tubes were centrifuged at 4 °C for 3 min at $15,000 \times g$ and the supernatant was discarded. The chromatin pellet was gently rinsed with cold 1× PBS/1 mM EDTA.

## Western blot

Different salt concentration treated chromatin and its supernatant were separately mixed with NuPAGE LDS Sample Buffer (NP0007, Invitrogen) and heated in a metal bath at 95 °C for 10 min. The ML-792-treated cells were harvested and lysed in RIPA buffer containing 1% EDTA-free Protease Inhibitor Cocktail (4693132001, Roche), mixed with NuPAGE LDS Sample Buffer and heated in a metal bath at 95 °C for 10 min. The sample proteins were separated by Tris-Acetate Precast PAGE Gel (P0538S, Beyotime) electrophoresis and transferred onto the polyvinylidene difluoride membranes (IPVH00010, Merck Millipore). The membranes were blocked with 5% BSA (V900933, Sigma) and incubated with the anti-SUMO2/3 antibody (M114-3, MBL, 1:2000), anti-CTCF antibody (3418S, CST, 1:2000), anti-H3K27me3 antibody (9733S, CST, 1:5000) and β-tubulin (T0100, Lablead, 1:10,000).

After washing, the membranes were further incubated with HRP-conjugated secondary antibodies at room temperature for 1 h. The blots were then detected with enhanced chemiluminescence (E1050, Lablead).

## X-ChIP experimental procedures

About $5-6 \times 10^6$ cells were crosslinked in 1% formaldehyde (F8775, Sigma) for 10 min at room temperature and quenched by adding 2.5 M glycine (V900144, Sigma) to a final concentration of 125 mM, washed in ice-cold PBS (P1022, Solarbio). Then the cells were resuspended in 1 mL of Lysis Buffer (5 mM PIPES, pH 8.0; 85 mM KCl; 0.5% NP-40; 10% glycerol) and rotated at 4 °C for 15 min. The nuclei were collected by brief centrifugation and resuspended in 200 μL of RIPA Buffer (1× PBS; 1% NP-40; 0.5% sodium deoxycholate; 0.1% SDS) and sonicated into 200–400 bp fragments using Bioruptor. Before Immunoprecipitation, the anti-Rabbit IgG antibody (2729S, CST, diluted in 0.5% BSA in PBS) and anti-CTCF antibody (diluted in 0.5% BSA in PBS) were incubated with pre-washed Dynabeads Protein A (10002D, Thermo Fisher) at 4 °C for 4–6 h or overnight. Of the sheared chromatin, 5% was set aside as Input, and the rest was immunoprecipitated with the Protein A-IgG (1:200) for 30 min at 4 °C and the Protein A-CTCF (1:200) for 4–6 h at 4 °C. The beads were sequentially washed five times with 1 mL of LiCl Wash Buffer (100 mM Tris-HCl pH 7.5; 500 mM LiCl; 1% NP-40; 1% sodium deoxycholate). After the washing steps, beads were incubated at 65 °C for 20 min with periodic gentle vortex in 200 μL of X-Elution Buffer (50 mM Tris-HCl, pH 8.0; 10 mM EDTA, pH 8.0; 1% SDS). 190 μL of X-Elution Buffer was added to the Input. The supernatant and Input were de-crosslinked at 65 °C for 6 h. Then, 200 μL of TECa buffer (10 mM Tris-HCl, pH 8.0; 1 mM EDTA, pH 8.0; 10 mM $CaCl_2$) and 10 μL of proteinase K (3115852001, Roche) were added to the reactions, which were incubated at 55 °C for 30 min. DNA was extracted by phenol:chloroform:isoamyl alcohol (A14140, OKA) extraction and ethanol precipitation. The ChIP-seq libraries were constructed using the following steps: (a) End Repair was accomplished by using the NEBNext End Repair Module (E6050L, NEB) at RT for 30 min. (b) A-tailing was accomplished by using 1 μL of *Taq* DNA Polymerase (EP0402, Thermo Fisher) and 3 μL of 10× A-tailing mix (100 μL of 10× ThermoPol Reaction Buffer (B9004S, NEB), 1 μL of 100 mM dATP) at 37 °C for 30 min. (c) Adapter ligation was accomplished by using 0.8 μL of T4 DNA Ligase (EL0012, Thermo Fisher), 0.6 μL of 45 μM Illumina adapters at room temperature for 2 h. PCR amplification was done with KAPA HiFi HotStart (KK2502, Roche) and was run using the following cycling condition: 95 °C 3 min for 1 cycle; 98 °C 30 s, 65 °C 30 s, and 72 °C 60 s for 4–8 cycles; and 72 °C 1 min for 1 cycle. PCR product was then cleaned-up by AMPure beads (18% PEG, 1:1). The libraries were sequenced using Illumina NovaSeq paired-end sequencing platform.

## N-ChIP experimental procedures

About $5-6 \times 10^6$ cells were collected by centrifugation at 4 °C for 3 min at $500 \times g$, washed with 1× PBS. After centrifugation, the pellet was resuspended in 1 mL of pre-chilled Lysis Buffer 1 (20 mM Tris-HCl pH 8.1; 150 mM NaCl; 3 mM $CaCl_2$; 0.1% Tween-20; 0.1% NP-40; 0.01% Digitonin) and incubated on ice for 3 min. Then 1 mL of Lysis Buffer 2 (20 mM Tris-HCl pH 8.1;

150 mM NaCl; 3 mM $CaCl_2$; 0.1% Tween-20) was added to the suspension and pre-warmed in 37 °C water bath for 3 min. 0.2 μL of Micrococcal Nuclease (MNase) (M0247S, NEB) was added to the tube and rotated in 37 °C water bath for 15 min. The reaction was stopped by 40 μL of 0.5 M EGTA (E3889, Sigma). 2 mL of liquid were evenly aliquoted into two new tubes and sonicated with 2–5 cycles using Bioruptor. Then the tubes were rotated at 4 °C for 20 min and centrifuged at 4 °C for 2 min at $15,000 \times g$. Before Immunoprecipitation (IP), the anti-Rabbit IgG antibody and anti-CTCF, anti-MAZ (ab85725, Abcam, diluted in 0.5% BSA in PBS), or anti-ZNF143 (16618-1-AP, Proteintech, diluted in 0.5% BSA in PBS) antibody were incubated with pre-washed Dynabeads Protein A at 4 °C for 4–6 h or overnight. Of the digested chromatin, 5% was set aside as Input, and the rest was immunoprecipitated with the Protein A-IgG for 30 min at 4 °C and the Protein A-CTCF (1:1000)/MAZ (1:1000)/ZNF143 (1:400) for 4–6 h at 4 °C. For CTCF, the concentration of NaCl was 75/150/225 mM; for MAZ and ZNF143, it was 50 mM. The beads were sequentially washed five times with 1 mL of Wash Buffer (20 mM Tris-HCl pH 8.1; X mM NaCl; 2 mM EDTA; 1% Triton X-100; 0.05% SDS). Chromatin complexes and Input were eluted by incubation with 300 μL of N-Elution Buffer (20 mM Tris-HCl pH 8.0; 10 mM EDTA; 5 mM EGTA; 0.1% SDS; 300 mM NaCl) and 10 μL of proteinase K at 55 °C for 30 min with periodic gentle vortex and supernatant was removed using a magnetic rack. DNA was extracted with phenol:chloroform and precipitated with ethanol. The AMPure beads (18% PEG, 1:1.4) were added to DNA solution. After 10 min, the supernatant was taken for ethanol precipitation to obtain small fragments of DNA. The N-ChIP-seq libraries were constructed using the following steps: (a) End repair was accomplished by using the NEBNext End Repair Module at RT for 30 min and DNA was cleaned-up by ethanol precipitation. (b) A-tailing was accomplished by using 1 μL of *Taq* DNA Polymerase and 3 μL of 10× A-tailing mix (100 μL of 10× ThermoPol Reaction Buffer, 1 μL of 100 mM dATP) at 37 °C for 30 min and DNA was cleaned-up by ethanol precipitation. The AMPure beads (18% PEG, 1:1.4) were added to DNA solution. After 10 min, the supernatant was taken for ethanol precipitation to obtain small fragments of DNA. (c) Adapter ligation was accomplished by using 0.8 μL of T4 DNA Ligase, 0.6 μL of 45 μM Illumina adapters at room temperature for 2 h and DNA was cleaned-up by AMPure beads (18% PEG, 1:1.4). PCR reaction was run using the following cycling condition: 95 °C 3 min for 1 cycle; 98 °C 20 s, 60 °C 15 s, and 72 °C 30 s for 6–8 cycles; and 72 °C 1 min for 1 cycle. PCR product was then cleaned-up by AMPure beads (18% PEG, 1:1.4). The libraries were sequenced using Illumina NovaSeq paired-end sequencing platform.

## Sequential N-ChIP experimental procedures

The experimental steps for harvesting fresh cells, lysis and MNase digestion were the same as N-ChIP procedures. Before Immunoprecipitation, the anti-Rabbit IgG antibody, anti-CTCF antibody and anti-SUMO2/3 antibody (diluted in 0.5% BSA in PBS) were incubated with pre-washed Dynabeads Protein A at 4 °C for 4–6 h or overnight. In the first round, chromatin was immunoprecipitated using the Protein A-IgG for 30 min at 4 °C and the Protein A-CTCF (1:1000) for 4–6 h at 4 °C. The beads were washed five times with 1 mL of Wash Buffer and resuspended in 75 μL TE

(10 mM Tris-HCl of pH 8.0; 2 mM EDTA)/10 mM DTT. The immuno-complexes were eluted by incubating 30 min at 37 °C and diluted 20 times (to a final volume of 1.5 mL) with Dilution Buffer (1% Triton X-100; 2 mM EDTA; 20 mM Tris-HCl of pH 8.1; 75 mM NaCl). Then the eluted chromatin was used for the second round IP with Protein A-SUMO2/3 (1:1000) for overnight at 4 °C and divided into the supernatant group (CTCF$^+$SUMO2/3$^-$) and the beads group (CTCF$^+$SUMO2/3$^+$). The beads were washed five times with 1 mL of Wash Buffer. The NaCl concentration in IP and wash buffer was 75 mM. The DNA extraction, size selection and sequencing libraries were prepared as described above.

## loMNase-seq experimental procedures

The experimental steps for harvesting fresh cells and lysis were the same as N-ChIP procedures. The difference was that MNase digestion time was 4 min. 30 μL of proteinase K was added to the chromatin complexes, which was incubated at 55 °C for 30 min. DNA was extracted through ethanol precipitation. The DNA size selection and sequencing libraries were prepared as described above.

## Micro-C experimental procedures

K562 cells were fixed by freshly made 1% formaldehyde in PBS for 10 min at room temperature. Crosslinking was then quenched with 0.125 M glycine for 5 min. Cells were rinsed twice with ice-cold PBS, then resuspend cell pellet in the long crosslinker Ethylene glycol-bis (succinicacid *N*-hydroxysuccinimide ester) (EGS) (E3257, Sigma) solution at a final concentration of 3 mM. Incubate for 45 min at room temperature with mixing. Add glycine to a final concentration of 0.4 M to quench the reaction. Incubate for 5 min at room temperature. Cells were rinsed again with ice-cold PBS. Cell pellets were then snap-frozen in liquid nitrogen and can be stored at −80 °C if necessary. Cells were lysed and nuclei were extracted. Nuclei were digested by MNase for 13 min. The digested DNA was blunt-ended by End Repair Mix, then A-tailing overhangs were added by *Taq* DNA Polymerase. The A-tailed DNA was ligated with a bridge linker (Forward: /5Phos/ GCCCGG/iBiodT/NNACGCCCGT, Reverse: /5Phos/CGGGCGTN NACCGGGCT) by T4 DNA ligase at 16 °C for 4 h. The excessive bridge linkers were removed by Lambda Exonuclease (M0262L, NEB) and Exonuclease I (M0293L, NEB). Proteinase K and 1% SDS was added and incubated at 65 °C overnight. DNA was purified by phenol:chloroform:isoamyl alcohol extraction and ethanol precipitation. The DNA was run on a 2.5% agarose gel, selected 250–400 bp region and gel-purified. DNA fragments with proximity linker ligation were enriched by C1 Streptavidin beads (65002, Thermo Fisher). Purified DNA was proceeded on streptavidin beads with end repair, A-tailing and ligation to Illumina adapters. The DNA was then subject to 6–10 cycles of PCR using Illumina paired-end primers and KAPA HiFi HotStart. Amplified library was purified and sequenced on Illumina NovaSeq X Plus platform.

## Public data

Public data used in this study, including MNase-seq, DNase-seq, ATAC-seq, X-ChIP, N-ChIP, CUT&Tag, CUT&RUN, ChIP-exo, in situ Hi-C, Micro-C, ChIA-PET, HiPore-C, RNA-seq and GRO-seq, were summarized in Dataset EV1.

## loMNase-seq, ATAC-seq, and DNase-seq data analysis

Raw reads are subjected to trim_galore (version 0.6.10, https://www.bioinformatics.babraham.ac.uk/projects/trim_galore/) to remove adapters and low-quality reads, then the trimmed reads were mapped to hg38 genome using bowtie2 (version 2.5.1) (Langmead and Salzberg, 2012). Reads aligned to the mitochondrial genome as well as reads with low mapping quality (MAPQ < 10) were discarded. Picard (version 2.25.7, https://broadinstitute.github.io/picard/) was used to mark and remove duplicates. BEDTools toolkit (version 2.30.0) (Quinlan and Hall, 2010) were used to create bedGraph files consisting of reads per million (RPM) values, then the bedGraph files were converted to bigWig file using UCSC bedGraphToBigWig (version 2.8) (Kuhn et al, 2013). Peaks for each replicate were called using MACS2 (version 2.2.7.1) (Zhang et al, 2008). All peaks that matched the ENCODE blacklist regions (Amemiya et al, 2019) of the genome were also filtered using BEDTools. ATAC-seq and DNase-seq data were downloaded from ENCODE (Dunham et al, 2012).

### Motif enrichment analysis

The top 3000 loMNase-seq peaks (sorted by signals) were extracted and expanded to −200 bp to +200 bp around the summits. DNA sequences of these regions were acquired using the BEDtools getfasta tool and used as input for de novo motif analysis with MEME-ChIP (version 5.4.1) (Machanick and Bailey, 2011). The newly discovered motifs were compared to the known motif database using TOMTOM (version 5.4.1) (Gupta et al, 2007). CentriMo (version 5.4.1) (Bailey and Machanick, 2012) was used to analyze the location of the de novo motifs over the 400 bp sequences. All tools are available at the MEME suite (http://meme-suite.org).

### Co-localization analysis

The processed narrow-Peak bed files for 709 K562 TF ChIP-seq datasets of 488 distinct TFs were collected from the ENCODE database. BEDtools intersectBed tool was used to determine the fraction of loMNase-seq peaks that are co-bound with specific TFs. Finally, the data was sorted based on the maximum overlap ratio for each TF.

## Motif detection for CTCF, MAZ, and ZNF143

CTCF motifs ever detected as occupied in at least one cell type was compiled from all available CTCF ChIP-seq experiments in human cell lines deposited in ENCODE as previously described (Xu and Corces, 2018).

MAZ binding regions was established by merging publicly available K562 DNase-seq, and MAZ ChIP-seq peaks with BEDtools. The DNA sequences of these regions and the MAZ position weight matrix (PWM) (MAZ_HUMAN.H11MO.1.A) obtained from the HOCO-MOCO (Kulakovskiy et al, 2018) database were fed to FIMO (Grant et al, 2011) to search for MAZ motifs, with parameters "--parse-genomic-coord --max-stored-scores 1000000 --thresh 1e-4".

ZNF143 motifs were detected in a similar manner except for the PWMs were generated as following: K562 ZNF143 ChIP-seq peaks present in at least three publicly available datasets (GEO: GSE180175 (Liu et al, 2022), GSE39263 (Ngondo-Mbongo et al, 2013). ENCODE: ENCSR000EGP, ENCSR427WZJ) were selected and their sequences was used for de novo motif discovery with MEME(version 5.4.1) (Bailey and Elkan, 1994) (-revcomp -dna

-nmotifs 3 -minw 6 -maxw 25 -mod zoops -nostatus) to obtain the top two statistically significant PWMs for ZNF143.

## V-plot visualization

To generate V-plots, the alignments from replicates were pooled. The length of each fragment was plotted as a function of the distance from the fragment midpoint to the center of the site for each annotated feature. Heatmaps were generated by R package ggplot2 (https://ggplot2.tidyverse.org/) (Wickham et al, 2016).

## ChIP-seq data analysis

Quality check of sequence libraries was performed by FastQC (version 0.11.8, https://www.bioinformatics.babraham.ac.uk/projects/fastqc/) with default parameters. Raw fastq reads were trimmed by trim_galore to remove any adapter sequences. Paired-end reads were shortened to 100 bp by Cutadapt (version 4.4) (Martin, 2011). The trimmed reads were aligned to the human reference genome hg38 by bowtie2 with options: --no-unal --no-discordant --no-mixed --very-sensitive --score-min L,0,-0.4 -X 1000. Bam files were filtered based on alignment quality (MAPQ ≥ 10) using SAMtools (version 1.6) (Li et al, 2009). Duplicates were removed by MarkDuplicates from Picard tools. For single-end ChIP-seq, the resulting alignments were extended to 100 bp form the 5' end. For visualization of all ChIP-seq as tracks, bamtobed and genomecov from BEDTools toolkit were used to create bedGraph files consisting of RPM values. The bedGraph files were converted to bigWig files for fast query retrievals using UCSC bedGraphToBigWig. ChIP-seq coverage tracks (for example, Fig. EV1B) were visualized using Integrative Genomics Viewer (version 2.8.10) (Robinson et al, 2011).

### Peak calling
Peaks for each replicate were called using MACS2 with the parameters "-keep-dup all --nomodel -q 0.05 -g hs". The Input sample for X-ChIP served as the control for peak calling, and N-ChIP did not normally require Input control (Kasinathan et al, 2014). All peaks that matched the ENCODE blacklist regions of the genome were also filtered using BEDTools.

### Motif annotation for CTCF (core, upstream, downstream)
To obtain optimal PWMs for the U-, C-, D- motif, the DNA sequences upstream, overlapping, and downstream of the C-motif contained in N1 sites were fed into MEME for de novo motif analysis. The R package ggseqlogo (Wagih, 2017) was used to drawing sequence logos, as shown in Fig. 3G. The SpaMo tool (version 5.4.1) (Whitington et al, 2011) was used for distance analysis between C-motifs and auxiliary motifs.

### Identification of typical enhancers and super enhancers
H3K27ac ChIP-seq data was used to identify typical enhancers and super enhancers in K562 cells by ROSE algorithm (Lovén et al, 2013; Whyte et al, 2013) (https://bitbucket.org/young_computation/rose). Briefly, enhancers were defined as H3K27ac ChIP-seq peaks identified using macs2. To identify super enhancers, the H3K27ac ChIP-seq peaks were merged if they were within 12.5 kb, and the merged enhancers were ranked based on the H3K27ac ChIP-seq signal. Regions within 2 kb from TSS were excluded. ROSE separates super enhancers from typical enhancers by identifying an inflection point of H3K27ac signal versus enhancer rank.

## Principal component analysis (PCA) for ChIP-seq and loMNase-seq samples

PCA was performed with the ChIPQC program (Carroll et al, 2014) with the parameters: annotation = "hg38", consensus=TRUE.

## Enrichment plots for ChIP-seq and loMNase-seq samples

Enrichment heatmaps and average profile plots (for example, Fig. 2B) were generated using the computeMatrix, plotProfile and plotHeatmap tools from deepTools suite (Ramírez et al, 2014).

## Overlap analysis

The ChIPpeakAnno package (version 3.24.2) from Bioconductor (Zhu et al, 2010) was used to draw Venn diagrams to visualize the overlap among loMNase-seq or ChIP-seq datasets. To eliminate the influence of sequencing depth, in Fig. 1A, the same number of alignments were utilized for each sample to identify peaks and perform overlap analysis. In addition, BEDTools was also used for the assessment of overlaps.

## Differential peak analysis

DiffBind R package (version 3.0.15) (Ross-Innes et al, 2012) was used to identify differential peaks for loMNase-seq or N-ChIP datasets, using DESeq2 (Love et al, 2014) method. Sites with an FDR < 0.05 and absolute $\log_2$ (fold change) ≥ 1 were defined as statistically significant.

## Derivation of Salt Tolerance Score (S-score)

The CTCF motifs were annotated by N-ChIP signals (biological replicates pooled) under 75, 150, or 225 mM NaCl concentration, respectively. The PCA was used to extracting binding features within these motifs by the "principal" function in the psych R package (version 2.1.9). The resulted PC1 values were defined as Salt Tolerance Score (S-score). The same analysis was conducted to establish the S-score for MAZ.

## S-score based grouping of TF motifs

TF motifs were grouped (for example, six groups of CTCF motifs in Appendix Fig. S2E), with each group exhibiting equal X-ChIP signal but different S-score. In brief, motifs with the same X-ChIP signal (and loMNase-seq signal) were individually selected and sorted based on S-score. Subsequently, the motifs within each specific set, which comprised motifs with the same X-ChIP signal, were assigned to different motif subsets according to their S-score from high to low.

## CUT&Tag and CUT&RUN data analysis

Published K562 CUT&Tag and CUT&RUN data were downloaded and re-analyzed. The analysis steps were almost the same as ChIP-seq, except for the bowtie2 options as previously described (Kaya-Okur et al, 2019; Meers et al, 2019): bowtie2 --no-unal --no-discordant --no-mixed --local --very-sensitive -I 10 -X 700 --phred33.

## Digestion boundary analysis

All alignments were oriented in the same direction relative to their nearest CTCF motif. The 5' and 3' ends of the alignments with the length ≤ 120 bp were normalized to RPM values to plot average profiles around CTCF motifs. The average profiles were then scaled by dividing the total average signals within the 200 bp region centered on the CTCF motifs.

## RNA-seq data analysis

Published K562 RNA-seq data were downloaded and reanalyzed. RNA-seq samples were aligned by HISAT2 (Kim et al, 2015) with parameters: --dta --no-unal --no-mixed --no-discordant --add-chrname --score-min L,0,-0.4 -X 1000. Bam files were filtered based on alignment quality (MAPQ ≥ 10) and used to calculate transcripts levels (transcripts per million (TPM)) of each gene through the StringTie (Pertea et al, 2015) software.

## GRO-seq data analysis

Published K562 GRO-seq data were downloaded and reanalyzed. Adapters were removed with trim_galore and then mapped to a single copy of the ribosomal gene DNA (rDNA) locus (GenBank accession #: U13369.1) to remove related transcribed sequences. Reads that did not map to rDNA were then mapped to human reference genome hg38. Bam files were filtered based on alignment quality (MAPQ ≥ 10) using SAMtools. Duplicates were removed by MarkDuplicates from Picard tools. Alignments were normalized to RPM values and Bigwig files were created using the bedGraphToBigWig tool.

## Cumulative percentage plot

CTCF motifs were annotated with S-score or normalized signals from CUT&RUN, CUT&Tag, or X-ChIP, then sorted separately in descending order based on each signal, resulting in individual rank values for each signal. The rank values of motifs containing certain features were extracted and divided by the total number of motifs. The cumulative percentage plots were produced with ggplot2 in R.

## DNA cyclizability analysis

DNAcycP software (Li et al, 2022) was used to predict intrinsic DNA cyclizability of DNA sequence around CTCF motifs.

## K562 genomic context analysis

RefSeq genes annotations for hg38 were downloaded from the UCSC Genome Browser at https://hgdownload.soe.ucsc.edu/goldenPath/hg38/bigZips/genes/. Promoter and genic regions were defined as 1 kb regions upstream from the transcriptional start sites to 0.5 kb regions downstream from the transcriptional end sites. The complement of all promoter and genic regions was defined as distal intergenic regions using BEDtools complement tool.

## K562 chromatin states analysis

The published histone modifications ("H3K27ac", "H3K27me3", "H3K36me3", "H3K4me1", "H3K4me2", "H3K4me3", "H3K9ac", "H3K9me3", "H4K20me1") and CTCF ChIP-seq datasets were used as input for ChromHMM (Ernst and Kellis, 2012) to define the chromatin states in K562 cells. This classification resulted in 11 distinct states, as shown in Appendix Fig. S1A.

## Micro-C data analysis

Adapter sequences and bridge linkers were trimmed using trim_galore and Cutadapt. All the resulting data were mapped and filtered using HiC-Pro analysis pipeline (version 3.1.0) (Servant et al, 2015) to obtain valid Micro-C contact read pairs. Duplicated pairs were removed by considering genomic position and UMI. Reproducibility analysis of Micro-C replicates was evaluated by 3DChromatin_ReplicateQC package (Yardımcı et al, 2019). Three independent algorithms, including GenomeDISCO (Ursu et al, 2018), QuASAR (Sauria and Taylor, 2017), and HiC-Rep (Yang et al, 2017), were used to calculate the reproducibility scores. Due to the good reproducibility of our data, all valid pairs from both replicates were merged. Output files containing all valid pairs were further converted to COOL files and HIC files using the "hicpro2higlass.sh" and "hicpro2juicebox.sh" in utilities of Hi-C-Pro, respectively. Contact matrices were then normalized using iterative correction (Imakaev et al, 2012) in COOL files or Knight-Ruiz (Knight and Ruiz, 2013) in HIC files. The HIC files were visualized with Juicebox (Durand et al, 2016a).

### PCA of Micro-C samples
All Micro-C loops detected under DMSO or TPA conditions were merged. The chromosight quantify algorithm (Matthey-Doret et al, 2020) was employed to respectively calculate loop scores for the merged loops using balanced COOL files derived from biological replicates of Micro-C experiments conducted under DMSO or TPA conditions. These loop scores were then used to generate principal components analysis plot for Micro-C samples.

### Interaction decay analysis
For *P(s)* plots and derivatives, the *cis* reads from the valid pairs files were used to calculate the contact probability (*P*) as a function of genomic separation *(s)* following the tutorial on cooltools GitHub page (https://github.com/open2c/cooltools). The corresponding derivative plot was calculated from the *P(s)* curve.

### Compartment analysis
We performed the compartment analysis of Micro-C data with cooltools and further visualized the compartment strength utilizing saddle plot implemented in cooltools.

### Contact domain and boundary analysis
The insulation score analysis from the cooltools package or arrowhead transformation analysis from the Juicer package (https://github.com/aidenlab/juicer) (Durand et al, 2016b) were used to identify contact domains along the diagonal at 10 kb resolution independently. Aggregated plots for domains were generated through the python package Coolpup.py program (Flyamer et al, 2020) where "--rescale --rescale_size 99 --local --features_format bed" options were used.

### Loop analysis
Mustache (Roayaei Ardakany et al, 2020) algorithms were used to identify loops for the Micro-C data. Loops were called with

balanced contact matrices at resolutions of 1 kb, 2 kb, 4 kb, 10 kb, and 20 kb using the calling options "-pt 0.1 -st 0.88 -oc 2". We then combined all loops at different resolutions. If an interaction was detected as a loop at different resolutions, we retained the precise coordinates in finer resolution and discarded the coarser resolution.

### Aggregated peak analysis (APA)

For APA to assess loop intensity, loops were centered and piled up on a 200 kb × 200 kb matrix with 5 kb resolution balanced data by cooltools package.

### Aggregated plots for CTCF sites and boundaries

The cooltools package was used to calculate aggregate observed-over-expected contact frequency maps (pileup maps) centered at CTCF sites or domain boundaries and bounded by a fixed flanking genomic distance. Pileup maps are centered on the main diagonal at each feature's midpoint.

## In situ Hi-C data analysis

Published in situ Hi-C were processed in the similar way as Micro-C. Contact domains were called using the Juicer tools Arrowhead (version 1.22.01) (Rao et al, 2014) algorithm at 10 kb resolution. Loops were called using Juicer tools HiCCUPS (version 1.22.01) (Rao et al, 2014) algorithm at 5, 10, and 25 kb resolutions.

## Positional enrichment plot

Each loop domain identified was divided into 100 bins and extra 10 bins were added to the start and end positions. The 1 Mb regions around the arrowhead boundaries was divided into 100 bins. The overlap between CTCF sites in each of the bins was scored. *Y* axis depicts the percentage of overlapping CTCF sites per 1000 loop or boundaries.

## Enrichment of CTCF sites at domain boundaries

The observed/expected enrichment was calculated using CTCF sites at the domain boundaries compared to random CTCF sites at the domain boundaries. The random CTCF sites were obtained with BEDTools shuffleBed.

## ChIA-PET analysis

CTCF ChIA-PET data was downloaded from ENCODE and reanalyzed with ChIA-PET2 (version 0.9.3) (Li et al, 2017) software (-m 1 -k 1 -C 4 -A ACGCGATATCTTATC -B AGTCAGATAA-GATAT -l 20 -t 20 -M "--nomodel -q 0.05 -B --SPMR --call-summits") using the hg38 genome.

## Multi-way interactions

The multi-way contacts were predicted from pairwise ChIA-PET loops using quick-cliques (https://github.com/darrenstrash/quick-cliques). Percentage of native and non-native sites in anchors of pairwise or multi-way contacts were calculated. We randomly repositioned all predicted three-way contacts on the same chromosome, while maintaining the distance between anchors. The predicted and random three-way contacts were then intersected with HiPore-C data by BEDtools intersectBed to verify the authenticity of our predicted multi-way contacts.

## Statistical analysis

All of the statistical details can be found in the figure legends. Differences were analyzed with several statistical tests, as indicated in figure legends. Error bars represent the mean ± standard deviation. No blinding was performed.

# Data availability

The datasets and computer code produced in this study are available in the following databases: IoMNase-seq data: Genome Sequence Archive HRA005744 (https://ngdc.cncb.ac.cn/gsa-human/browse/HRA005744); X-ChIP/N-ChIP data: Genome Sequence Archive HRA005744 (https://ngdc.cncb.ac.cn/gsa-human/browse/HRA005744); Micro-C data: Genome Sequence Archive HRA005744 (https://ngdc.cncb.ac.cn/gsa-human/browse/HRA005744); Data analysis scripts: GitHub (https://github.com/husiling540/native_chromatin).

The source data of this paper are collected in the following database record: biostudies:S-SCDT-10_1038-S44320-024-00038-5.

# Peer review information

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

## Acknowledgements

This work was supported by grants from the Ministry of Science and Technology of China (National Key R&D Program of China: 2020YFA0803401 and 2022YFC2703303) and the National Natural Science Foundation of China (32070611 and 32370624).

## Author contributions

**Siling Hu**: Conceptualization; Data curation; Formal analysis; Investigation; Visualization; Writing—original draft; Writing—review and editing. **Yangying Liu**: Conceptualization; Investigation; Methodology; Writing—original draft; Writing—review and editing. **Qifan Zhang**: Investigation; Methodology. **Juan Bai**: Investigation. **Chenhuan Xu**: Conceptualization; Supervision; Funding acquisition; Investigation; Methodology; Writing—original draft; Writing—review and editing.

Source data underlying figure panels in this paper may have individual authorship assigned. Where available, figure panel/source data authorship is listed in the following database record: biostudies:S-SCDT-10_1038-S44320-024-00038-5.

## Disclosure and competing interests statement

The authors declare no competing interests.

# Expanded View Figures

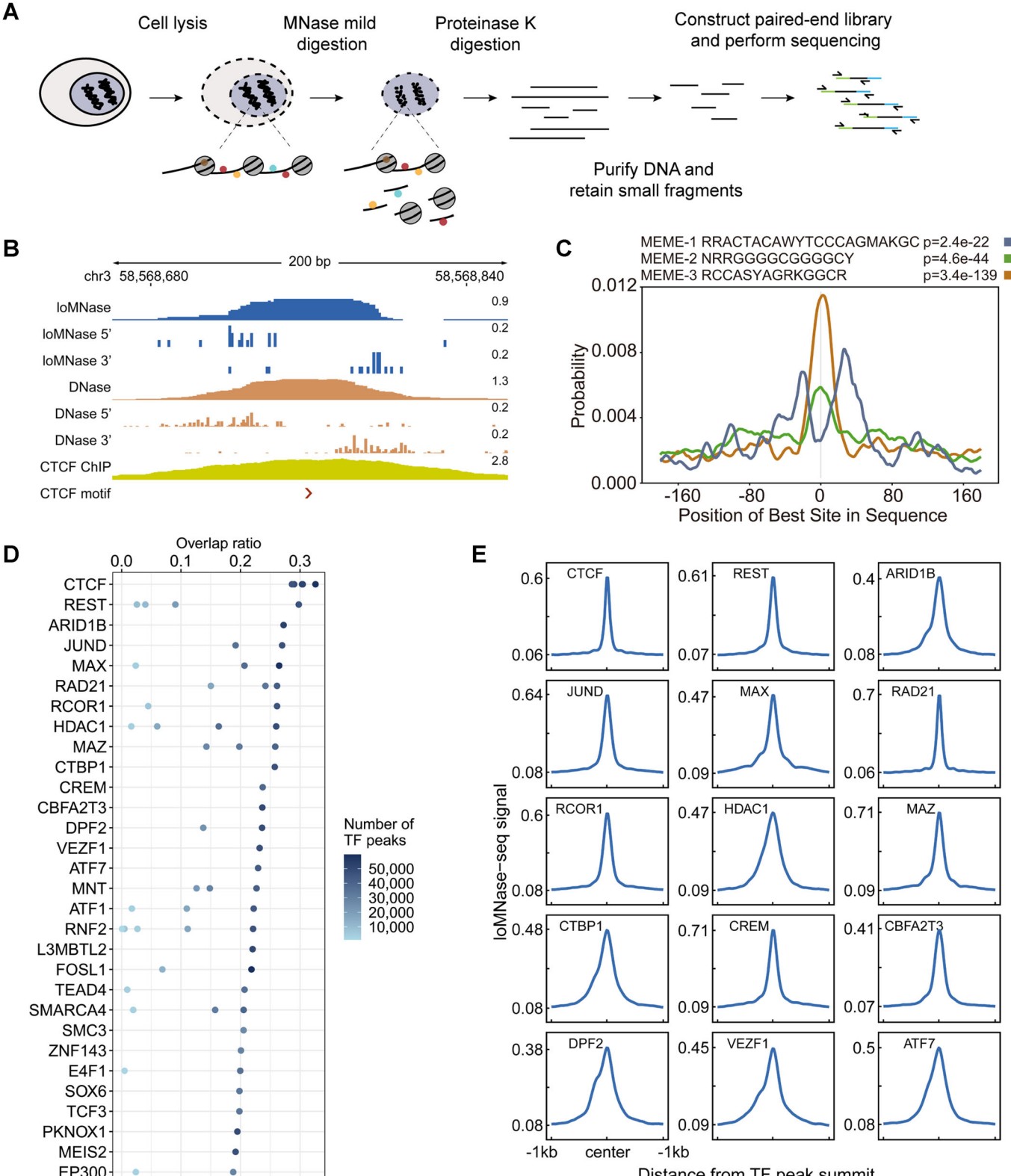

**Figure EV1.  loMNase-seq maps high-resolution TF footprints on native chromatin.**

(A) Schematic workflow of loMNase-seq. (B) Genome browser representation of normalized reads for loMNase-seq, DNase-seq, and CTCF ChIP-seq, as well as the fragment endpoints of loMNase-seq and DNase-seq relative to an example CTCF motif. CTCF motif orientation is indicated by a red arrow. (C) The CentriMo plot showing the distribution of the three most optimal MEME-ChIP-derived motifs in the top 3000 loMNase-seq peak regions. The centrally positioned vertical line corresponds to the midpoint of the sequences. MEME-1, -2, -3 correspond to Motif 1, 2, 3 respectively shown in Fig. 1E. The *P* value derived from CentriMo tool represents the statistical significance of the enrichment of the motif, adjusted for multiple tests. The *P* value is calculated by one-tailed binomial test. (D) The top 30 TFs were ranked based on their maximum overlap ratio with loMNase-seq peaks. Each dot represents a ChIP-seq result for a specific TF obtained from ENCODE database. (E) Enrichment of loMNase-seq signals at different TF ChIP-seq peak sets.

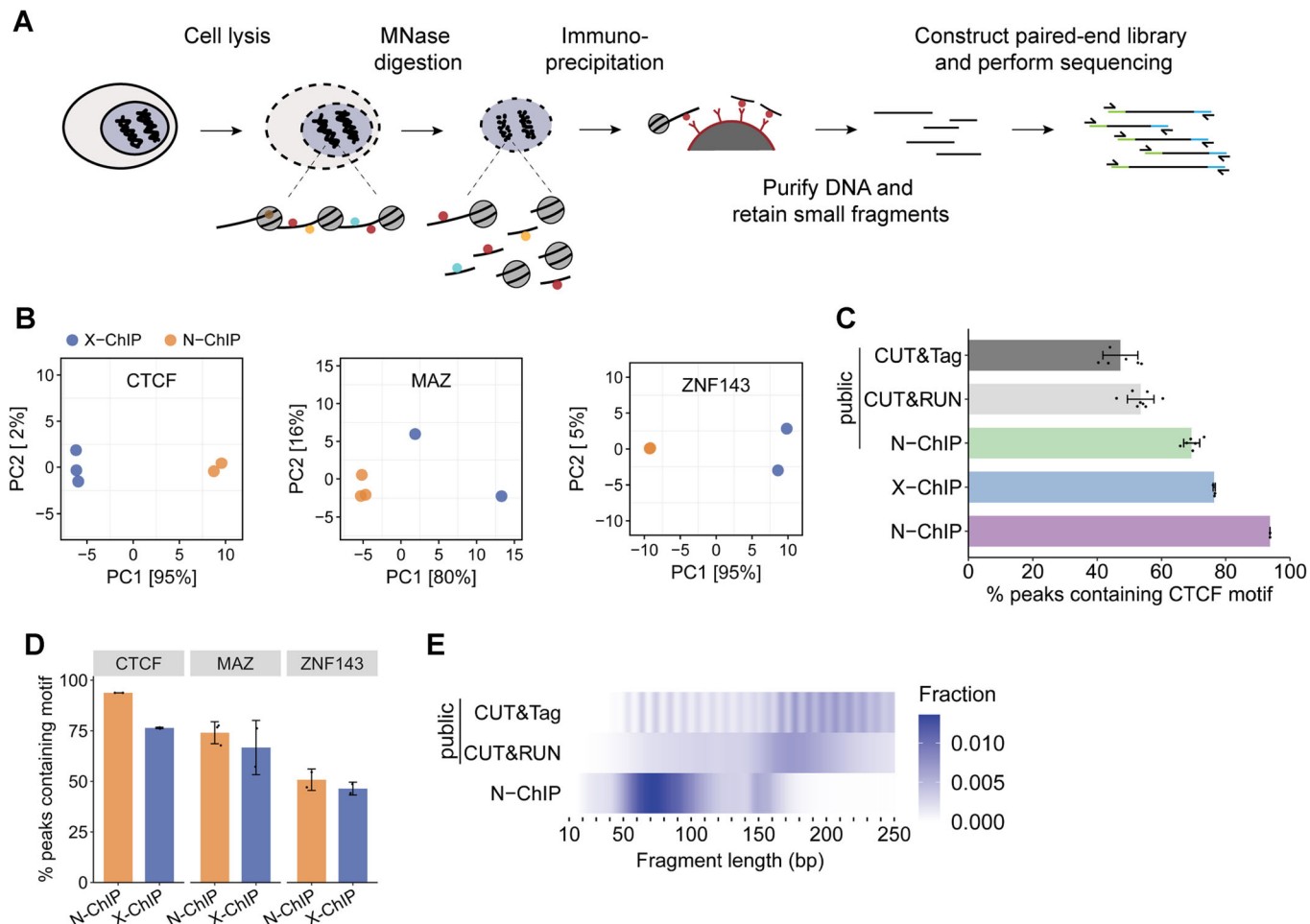

**Figure EV2.  Occupancy of single TFs on native chromatin.**

(**A**) Schematic workflow of N-ChIP. (**B**) Principal component analysis (PCA) of N-ChIP and X-ChIP datasets for CTCF, MAZ or ZNF143. Each dot represents an individual replicate. (**C**) Percentage of peaks containing CTCF canonical motifs for previously published CUT&Tag, CUT&RUN, N-ChIP, as well as N-ChIP and X-ChIP datasets generated in this study. Each dot represents an individual experiment or replicate. Numbers of samples (from top to bottom): $n = 6/8/6/6/3/2$. Error bars represent the mean ± standard deviation. (**D**) Percentage of CTCF, MAZ or ZNF143 peaks containing their respective motifs for N-ChIP or X-ChIP datasets. Numbers of samples (from left to right): $n = 2/3/3/2/2/2$. Error bars represent the mean ± standard deviation. (**E**) Fragment length distribution of CUT&Tag, CUT&RUN or N-ChIP datasets.

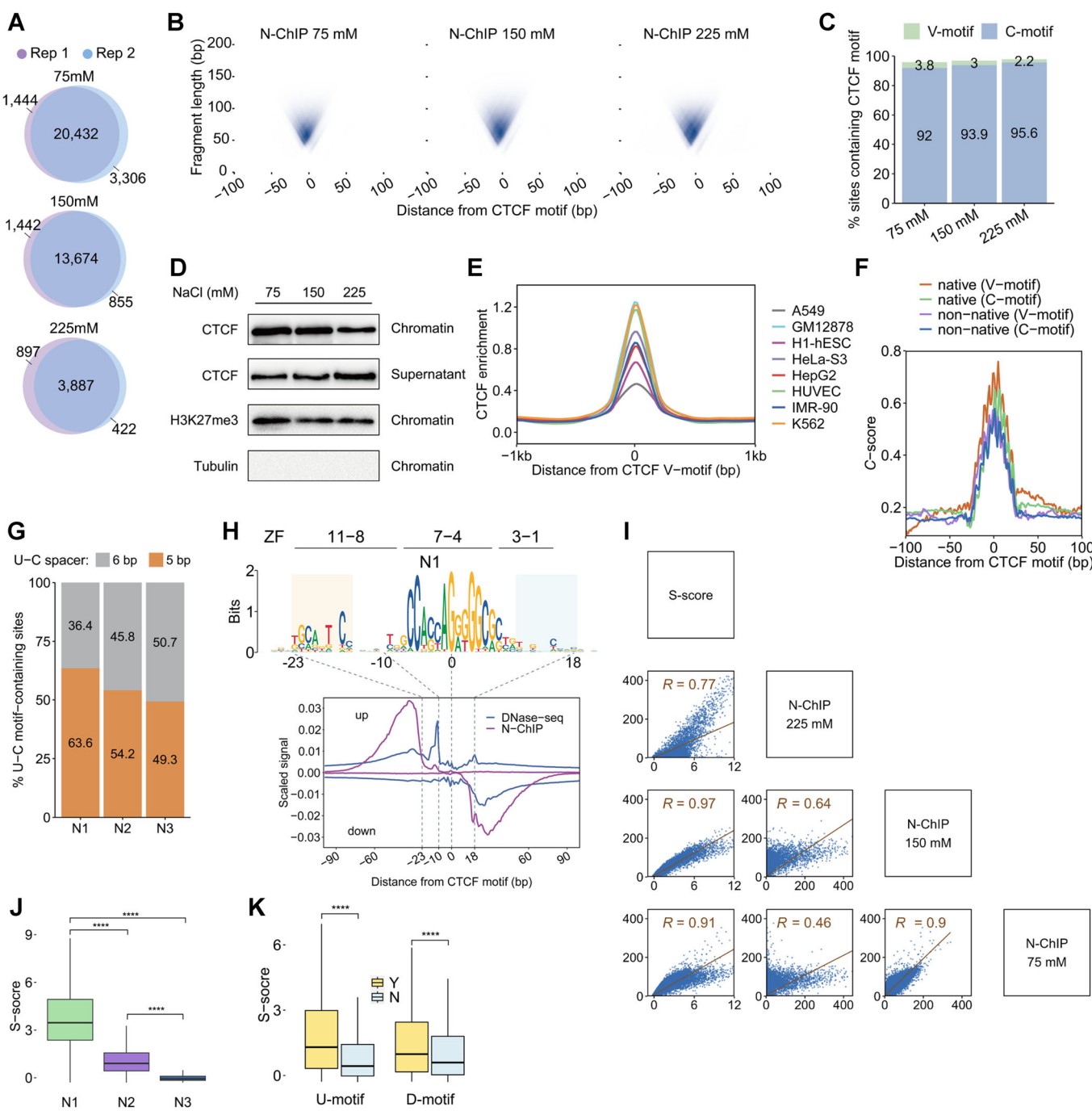

**Figure EV3.  A continuum of CTCF retention across genomic sites on native chromatin.**

(A) Venn diagrams showing reproducibility between biological replicates of CTCF N-ChIP under 75, 150, or 225 mM NaCl concentration. (B) V-plots of 75, 150, or 225 mM CTCF N-ChIP fragments at CTCF motifs. (C) Percentage of 75, 150, or 225 mM CTCF N-ChIP peaks containing C- or V-motifs. (D) Western blot of extracted chromatin fractions. The chromatin and supernatant on right side mark the insoluble and soluble fractions respectively, after incubating the extracted chromatin fraction in 75, 150, or 225 mM NaCl. H3K27me3 and Tubulin serve as controls for chromatin and cytoplasm respectively. (E) The normalized CTCF X-ChIP signal of eight different cell types at CTCF V-motifs. (F) Mean predicted cyclizability score (*C*-score) around C- or V-motifs present in native or non-native sites. (G) The percentage of CTCF motifs with different U-C spacer length from N1, N2 and N3 groups. (H) The scaled pileup of fragment 5' and 3' ends around CTCF motifs. Motif logo is shown at the top. The ZF domains binding to these regions are indicated. (I) Pairwise scatterplots showing the Pearson correlation between S-score and normalized N-ChIP signals. (J) Boxplot showing the S-score of CTCF motifs within N1, N2 and N3 sites (N1, $n = 3669$; N2, $n = 8823$; N3, $n = 6304$). ****$P < 0.0001$; two-sided Wilcoxon rank sum test. (K) Comparison of S-score of N1, N2 and N3 C-motifs with (Y) or without (N) auxiliary motifs. Numbers of C-motifs (from left to right): $n = 5632/13,164/2032/16,764$. ****$P < 0.0001$; two-sided Wilcoxon rank sum test. Data information: For boxplots in (J, K), the central band represents the median. The lower and upper hinges represent the first and third quartiles, respectively. The whiskers represent the 1.5× interquartile range.

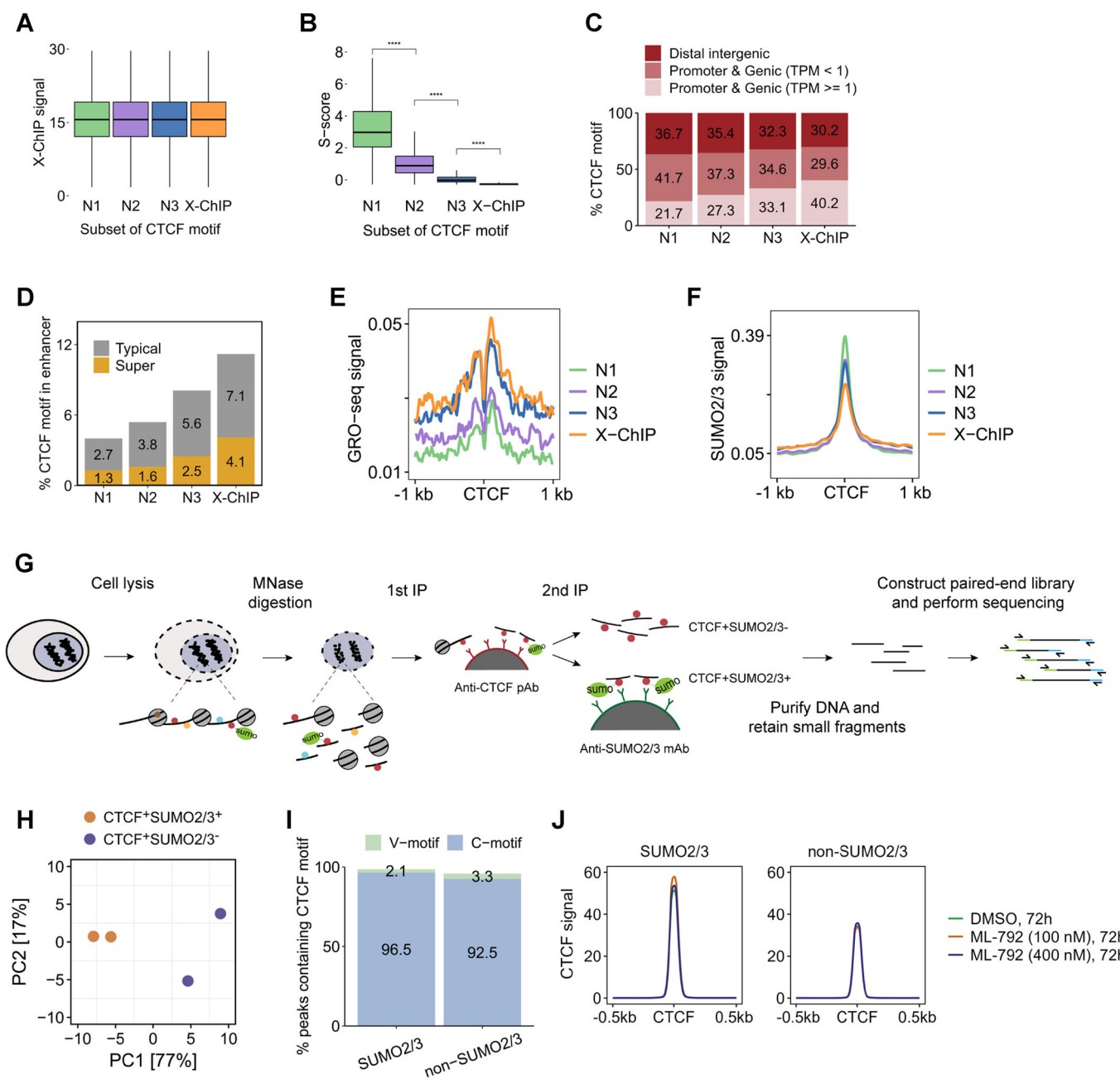

**Figure EV4. CTCF retention on native chromatin associates with transcriptional activity and post-translational modifications.**

(A) Boxplot showing the X-ChIP signals of sampled CTCF motifs from N1, N2, N3 and X-ChIP groups. For each subset, the number of CTCF motifs is 1,939. (B) Boxplot showing the S-score of sampled CTCF motifs from N1, N2, N3 and X-ChIP groups. For each subset, the number of CTCF motifs is 1,939. ****$P < 0.0001$; two-sided Wilcoxon rank sum test. (C) Percentage of sampled CTCF motifs from N1, N2, N3 and X-ChIP groups located within distal intergenic, or promoter and genic (TPM < 1 or ≥ 1) regions. (D) Percentage of sampled CTCF motifs from N1, N2, N3 and X-ChIP groups located within typical or super enhancers. (E) GRO-seq signals at sampled CTCF motifs from N1, N2, N3 and X-ChIP groups. (F) SUMO2/3 X-ChIP signal at sampled CTCF motifs from N1, N2, N3 and X-ChIP groups. (G) Schematic workflow of sequential N-ChIP. (H) PCA of CTCF-SUMO2/3 sequential N-ChIP datasets. Each dot represents an individual replicate. (I) Percentage of SUMO2/3 and non-SUMO2/3 sites containing C- or V-motifs. (J) Enrichment of CTCF N-ChIP signals around CTCF motifs within SUMO2/3 and non-SUMO2/3 sites under SUMOylation inhibition. Data information: For boxplots in (A, B), the central band represents the median. The lower and upper hinges represent the first and third quartiles, respectively. The whiskers represent the 1.5× interquartile range.

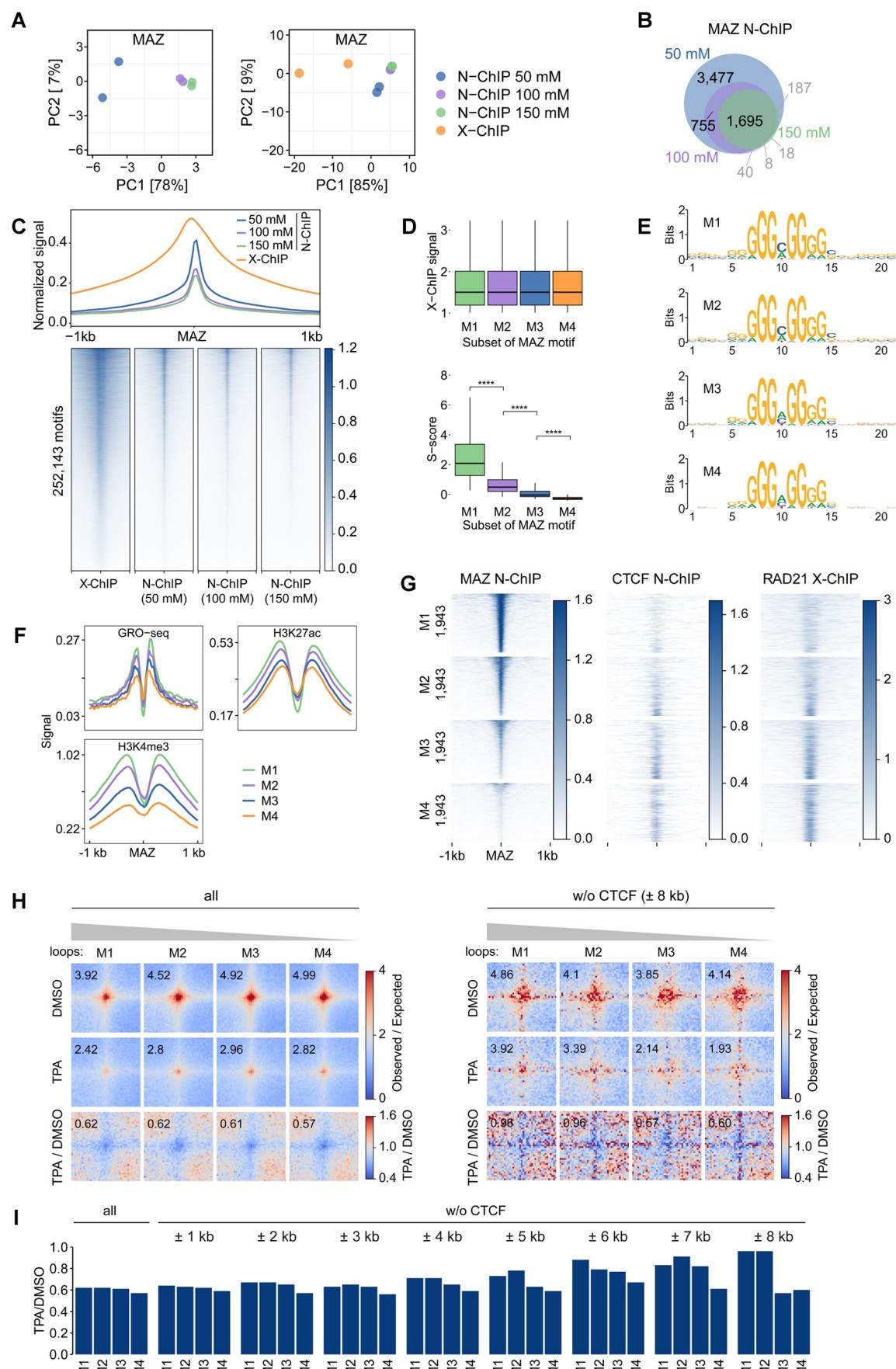

◀  **Figure EV5.  The S-score delineates MAZ retention on native chromatin and stability of MAZ-mediated chromatin loops independent of CTCF.**

(A) Principal component analysis of MAZ N-ChIP datasets with (right) or without (left) X-ChIP datasets. Each dot represents an individual replicate. (B) Venn diagram showing the overlap between 50, 100, and 150 mM MAZ N-ChIP peaks. (C) Average profiles and heatmaps showing normalized MAZ X-ChIP and N-ChIP signals centered around the MAZ motifs. (D) The boxplots showing the X-ChIP signals and S-score of four groups of MAZ motifs. The four groups of MAZ motifs are grouped based on identical X-ChIP signals (see "Methods"). For each subset, the number of MAZ motifs is 1943. The central band represents the median. The lower and upper hinges represent the first and third quartiles, respectively. The whiskers represent the 1.5× interquartile range. ****$P < 0.0001$; two-sided Wilcoxon rank sum test. (E) Sequence logo representation for the four groups of MAZ motifs as described in (D). (F) Average profiles showing GRO-seq signals and ChIP signals of different histone modifications centered around the four groups of MAZ motifs as described in (D). (G) Heatmaps showing MAZ N-ChIP (50 mM), CTCF N-ChIP (75 mM) and RAD21 X-ChIP signals centered around the four groups of MAZ motifs as described in (D). (H) APA plots showing the changes of chromatin loop strength under TPA treatment. The loops are grouped based on the presence of MAZ motifs from the four groups at their anchors, as defined in (D). Right panel: same as left panel but Micro-C pairs with CTCF motifs within ±8 kb were excluded to diminish contribution by CTCF-mediated loops. Bin size: 5 kb. (I) Barplot showing the fold change of chromatin loop strength between the TPA and DMSO conditions. Micro-C pairs with CTCF motifs within ±1, ±2, ±3, ±4, ±5, ±6, ±7 or ±8 kb were consecutively excluded to diminish the contribution by CTCF-mediated loops.

