## [Peer Review File · Molecular Systems Biology]

A continuum of zinc finger transcription factor retention on native chromatin underlies dynamic genome organization

Chenhuan Xu, Siling Hu, Yangying Liu, Qifan Zhang, and Juan Bai

Corresponding author(s): Chenhuan Xu (xuchh@big.ac.cn)

Review Timeline:

Submission Date:	27th Nov 23
Editorial Decision:	10th Jan 24
Revision Received:	14th Mar 24
Editorial Decision:	9th Apr 24
Revision Received:	10th Apr 24
Accepted:	15th Apr 24

Editor: Poonam Bheda

Transaction Report:

10th Jan 2024

Manuscript Number: MSB-2023-12136-T

Title: A continuum of transcription factor retention on native chromatin underlies dynamic genome organization

Dear Dr. Xu,

Thank you again for submitting your work to Molecular Systems Biology. We have now heard back from the three reviewers who agreed to evaluate your study. As you will see below, the reviewers appreciate that the proposed approach addresses a timely topic. However, they raise a series of concerns, which we would ask you to address in a major revision.

Without repeating all the comments listed below, editorially we would ask you to address the following points:

- In line with comments from Reviewer 2, the reproducibility of the method should be clear. Missing negative controls and spike-in normalization should also be included
- In line with comments from Reviewer 3, editorially we would encourage you to test your loMNase method and analysis on other more classical and/or labile TF(s) to clarify and highlight the utility of the method.

All other issues raised would need to be satisfactorily addressed. Please let me know in case you would like to discuss in further detail any of the comments, I would be happy to schedule a call.

We require:

- 1) A .docx formatted version of the manuscript text (including legends for main figures, EV figures and tables). Please make sure that the changes are highlighted to be clearly visible. Alternatively you may choose to submit your manuscript as a LaTeX file.
 - 2) Individual production quality figure files as .eps, .tif, .jpg (one file per figure). For guidance, download the 'Figure Guide PDF' (<https://www.embopress.org/page/journal/17574684/authorguide#figureformat>).
 - 3) At EMBO Press we ask authors to provide source data for the main figures. Our source data coordinator will contact you to discuss which figure panels we would need source data for and will also provide you with helpful tips on how to upload and organize the files.
 - 4) A .docx formatted letter INCLUDING the reviewers' reports and your detailed point-by-point responses to their comments. As part of the EMBO Press transparent editorial process, the point-by-point response is part of the Review Process File (RPF), which will be published alongside your paper.
 - 5) A complete author checklist, which you can download from our author guidelines (<https://www.embopress.org/page/journal/17574684/authorguide#submissionofrevisions>). Please insert information in the checklist that is also reflected in the manuscript. The completed author checklist will also be part of the RPF.
 - 6) Please note that all corresponding authors are required to supply an ORCID ID for their name upon submission of a revised manuscript.
 - 7) It is mandatory to include a 'Data Availability' section after the Materials and Methods. Before submitting your revision, primary datasets produced in this study need to be deposited in an appropriate public database, and the accession numbers and database listed under 'Data Availability'. Please remember to provide a reviewer password if the datasets are not yet public (see <https://www.embopress.org/page/journal/17574684/authorguide#dataavailability>).
- In case you have no data that requires deposition in a public database, please state so in this section. Note that the Data Availability Section is restricted to new primary data that are part of this study.
This study includes no data deposited in external repositories.
- 8) For data quantification: please specify the name of the statistical test used to generate error bars and P values, the number (n) of independent experiments (specify technical or biological replicates) underlying each data point and the test used to calculate p-values in each figure legend. The figure legends should contain a basic description of n, P and the test applied. Graphs must include a description of the bars and the error bars (s.d., s.e.m.). Please provide exact p values.
 - 9) Our journal encourages inclusion of *data citations in the reference list* to directly cite datasets that were re-used and obtained from public databases. Data citations in the article text are distinct from normal bibliographical citations and should

directly link to the database records from which the data can be accessed. In the main text, data citations are formatted as follows: "Data ref: Smith et al, 2001" or "Data ref: NCBI Sequence Read Archive PRJNA342805, 2017". In the Reference list, data citations must be labeled with "[DATASET]". A data reference must provide the database name, accession number/identifiers and a resolvable link to the landing page from which the data can be accessed at the end of the reference. Further instructions are available at .

<https://www.embopress.org/page/journal/17574684/authorguide#expandedview>

11) For more information: There is space at the end of each article to list relevant web links for further consultation by our readers. Could you identify some relevant ones and provide such information as well? Some examples are patient associations, relevant databases, OMIM/proteins/genes links, author's websites, etc...

12) Author contributions: CRediT has replaced the traditional author contributions section because it offers a systematic machine readable author contributions format that allows for more effective research assessment. Please remove the Authors Contributions from the manuscript and use the free text boxes beneath each contributing author's name in our system to add specific details on the author's contribution. More information is available in our guide to authors.

13) Disclosure statement and competing interests: We updated our journal's competing interests policy in January 2022 and request authors to consider both actual and perceived competing interests. Please review the policy <https://www.embopress.org/competing-interests> and update your competing interests if necessary.

14) Every published paper now includes a 'Synopsis' to further enhance discoverability. Synopses are displayed on the journal webpage and are freely accessible to all readers. They include a short stand first (maximum of 300 characters, including space) as well as 2-5 one-sentences bullet points that summarizes the paper. Please write the bullet points to summarize the key NEW findings. They should be designed to be complementary to the abstract - i.e. not repeat the same text. We encourage inclusion of key acronyms and quantitative information (maximum of 30 words / bullet point). Please use the passive voice. Please attach these in a separate file or send them by email, we will incorporate them accordingly.

Please also suggest a striking image or visual abstract to illustrate your article as a PNG file 550 px wide x 300-600 px high. Share synopsis text and image, as well as eTOC:

Please note that these would be the final versions and changes during proofing are usually not allowed

15) As part of the EMBO Publications transparent editorial process initiative (see our Editorial at <http://embomolmed.embopress.org/content/2/9/329>), Molecular Systems Biology Medicine will publish online a Review Process File (RPF) to accompany accepted manuscripts.

In the event of acceptance, this file will be published in conjunction with your paper and will include the anonymous referee reports, your point-by-point response and all pertinent correspondence relating to the manuscript. Let us know whether you agree with the publication of the RPF and as here, if you want to remove or not any figures from it prior to publication.

Molecular Systems Biology has a "scooping protection" policy, whereby similar findings that are published by others during review or revision are not a criterion for rejection. Should you decide to submit a revised version, I do ask that you get in touch after three months if you have not completed it, to update us on the status.

I look forward to receiving your revised manuscript.

Yours sincerely,

Poonam Bheda

Poonam Bheda, PhD
Scientific Editor

Reviewer #1:

In this work, the authors present two related techniques, loMNase-seq and N-ChIP, to assess transcription factor occupancy and genome organization on native (not crosslinked) chromatin. By utilizing a shortened digestion time, loMNase-seq preferentially identifies shorter DNA fragments, which results in high resolution of TF binding sites. By pairing this technique with subsequent ChIP for a TF of interest (primarily CTCF, but also MAZ and ZNF143), they identify a subset of the binding sites identified on traditional cross-linked ChIP assays, which can be further winnowed down by repeating their experiment with increasing salt concentrations, and results in a ranking of sorts of CTCF binding site strength. Subsequently, the authors correlate a variety of additional characteristics with CTCF binding site strength (e.g. presence of a canonical CTCF binding site motif, presence of an upstream U-motif, association with transcriptional activity, participation in a CTCF loop, etc.).

I think the manuscript is excellent and only have minor suggestions.

1. Throughout the paper, the authors use the S-score to demonstrate improved performance over X-ChIP as a means of ranking CTCF binding site strength and identifying associations with the above mentioned characteristics of these sites. However, the improvement seems rather marginal, and in Figure 3C in particular it appears that the stronger binding sites observed in the salt N-ChIP experiments, and their subsequent ranking by the S-score, could have been mostly predicted simply by identifying the X-ChIP peaks with the strongest signal. This has the negative effect of discounting the utility of the S-score, but it does raise an important point. It was not, I don't believe, self-evident that the peaks with the strongest signal on X-ChIP would correlate to those most resistant to salt (or RNAi, Zn, etc.). There are alternative explanations for a strong X-ChIP signal (e.g. a very high percentage of cells in the population have that particular site bound, even if weakly so). I think more should be made of this point as it is in my view a larger advantage to the technique than what is currently emphasized.
2. Could the authors comment on the rationale for the different proteinase K used in loMNase-seq versus sequential N-ChIP?
3. In Figure 2A, the gained peaks, particularly for ZNF143, are very interesting. I would be very surprised if epitope masking explains such a large number of peaks for ZNF143. The presence of gained peaks at all does not align with the author's framework of utilizing native techniques to avoid 'over-calling' of peaks by using formaldehyde. It would be very interesting to look further into the characteristics of these sites. I am also curious if CTCF is an outlier here without a significant number of gained peaks or not, and would like to see the data for more transcription factors than just these three.
4. The authors introduce a previously published N-ChIP data set in Figure 2C. As the development of a new methodology is a significant result of this paper, I would like to have more discussion about what the current method improves over previous methods performing occupancy assays on native chromatin, like the one introduced here.
5. Figure 6C should not be a line plot because the x-axis is categorical.
6. Additional commentary on the mechanism by which SUMOylation of CTCF influences its occupancy would add to the story.
7. I have concerns about comparing CTCF binding sites between different cell types (K562 versus LNCaP), although in their defense they did get good overlap of RNAi- and salt-resistant sites. It would be nice to see the heat maps to see if it is the 'most' salt-resistant sites (persist at highest salt concentration) that are also the RNAi-resistant sites.

Reviewer #2:

In this manuscript, the authors optimize a low MNase native method that can enrich for small fragments, and then use this in NChIP to look at native factor binding. The authors focus on CTCF and find CTCF dynamics in native chromatin, showing how increasing salt will remove a subset of CTCF bound sites and observe correlation with SUMOylation levels. Authors also examine TPA treatment (zinc depletion) and observe differential binding and microC profiles. Together, this manuscript has many interesting advancements, although many findings are largely descriptive.

- 1) With loMNase-seq, it is hugely surprising to me that there are no nucleosome size reads, as I would have anticipated that even without crosslinking the majority of MNase digested chromatin would result in nucleosomes and not only TF footprints. I do not have a question per se, but I would request that the authors offer an explanation to the lack of nucleosome sized reads.
- 2) It is unclear to me in the analysis if the authors do computational size selection for any of the analysis performed?
- 3) Are experiments spike in normalized? For many results there are global losses or reductions in protein occupancy and therefore it is important to establish spike in normalization for global changes.
- 4) The authors present a lot of data but the reproducibility is not clear to me. How many replicates for experiments and how robust are the results (present PCA plots or even better XY scatterplots for replicates).
- 5) In Figure 1, the authors present global motifs from loMNase-seq. Can the authors get more from these data, for example perform peak calling for only promoter locations? Or look over other ChIPseq datasets (for any factors of interest) for intensity of reads?
- 6) On lines 113-115 the two statements seem contradictory to me.
- 7) For data presented in Fig 2 and beyond for NChIP experiments, I do not see any negative controls (no antibody or IgG, for

example). Were these included to show background and to subtract for peak calling etc?

8) In Figure 3, the authors present progressive loss of CTCF from chromatin in increasing salt conditions. It would be great to show these results using biochemical assays, for example doing chromatin fractionation to examine whether CTCF is lost from chromatin using this orthogonal assay.

9) The authors data presented in Fig 4 suggest that SUMOylation on CTCF might facilitate retention of chromatin, and these is anticorrelated with transcription. It would be of great interest to investigate this hypothesis. For example, could the authors mutate the residues on CTCF to prevent SUMOylation and then test binding?

10) I think that Figure 5 is a very powerful figure, in terms of the retention and conservation across cell lines.

Reviewer #3:

In this manuscript, Hu and colleagues use a low-MNase-concentration variant of native (N) ChIP to look at presumed retention times of TFs on chromatin. Overall, the manuscript is well written, the experiments well explained, and the development of the S-score metric is really clever and potentially very useful. However, I have a couple of important conceptual issues with how the manuscript is presented. First, the use of low MNase tigers for performing native ChIP not new -- many labs will titrate their MNase to suit their target, and the usefulness of low MNase treatments for uncovering labile TF binding footprints has been exemplified before (e.g., in work by the Laengst lab). Second, footprinting aside, the actual N-ChIP data focus on three ZF TFs known for their strong chromatin association. Even more so, the vast majority of work in the paper only analysis CTCF, arguably the most stably-associated TF on chromatin. This puts into question the usefulness of the whole approach for any other TF (like p65, p53, etc.) that are notoriously labile in the chromatin association, not to mention proteins more promiscuous binding nature like HMGBs. Therefore, I would suggest that there are two avenues the authors can take (in my opinion): either provide data from a more "classical" TF to broaden the scope of the approach and show how their S-score performs in this case or thoroughly refocus the existing data to describe all their nice CTCF-related findings and refrain from generalisations that are not backed by data. In either case, following the suggested changes, the manuscript would be a very nice contribution to the field.

Secondary remarks:

Intro: the statement of crosslinking equally stabilizing short- and long-lived TF binding events in the course of 10 min is exaggerated and should be rephrased. Shorter-lived events will consistently have a lower probability of being stabilised and therefore will represent lower abundance/signal intensity events in a ChIP experiment.

Results: there are a lot of average plots shown but not a single actual browser view provided so as to be able to assess the signal structure in the N-ChIP experiments. The same applies to Micro-C data, which makes it hard to really assess the TPA treatment effects.

A. Papantonis

We thank the reviewers for the thoughtful and helpful comments. We have prepared an improved revised manuscript according to these comments. Below we provide a point-by-point response to address each of the reviewer's concerns.

New figures:

Figure 7C. The following figures in Figure 7 renumbered thereafter.

Figure EV1E.

Figure EV3D. The following figures in Figure EV3 renumbered thereafter.

Figure EV4J.

Figure EV6A, B, F, G. The other figures in Figure EV6 renumbered accordingly.

Figure EV7A, B. The other figures in Figure EV7 renumbered thereafter.

Figure EV8.

Changes to old figures:

Old Figure 3C, now new Figure 3C.

Old Figure 6C, now new Figure 6C.

Reviewer #1:

In this work, the authors present two related techniques, loMNase-seq and N-ChIP, to assess transcription factor occupancy and genome organization on native (not crosslinked) chromatin. By utilizing a shortened digestion time, loMNase-seq preferentially identifies shorter DNA fragments, which results in high resolution of TF binding sites. By pairing this technique with subsequent ChIP for a TF of interest (primarily CTCF, but also MAZ and ZNF143), they identify a subset of the binding sites identified on traditional cross-linked ChIP assays, which can be further windowed down by repeating their experiment with increasing salt concentrations, and results in a ranking of sorts of CTCF binding site strength. Subsequently, the authors correlate a variety of additional characteristics with CTCF binding site strength (e.g. presence of a canonical CTCF binding site motif, presence of an upstream U-motif, association with transcriptional activity, participation in a CTCF

loop, etc.).

I think the manuscript is excellent and only have minor suggestions.

Response: We thank the reviewer for summarizing the merits of our work, and for the nice comments on our findings.

1. Throughout the paper, the authors use the S-score to demonstrate improved performance over X-ChIP as a means of ranking CTCF binding site strength and identifying associations with the above mentioned characteristics of these sites. However, the improvement seems rather marginal, and in Figure 3C in particular it appears that the stronger binding sites observed in the salt N-ChIP experiments, and their subsequent ranking by the S-score, could have been mostly predicted simply by identifying the X-ChIP peaks with the strongest signal. This has the negative effect of discounting the utility of the S-score, but it does raise an important point. It was not, I don't believe, self-evident that the peaks with the strongest signal on X-ChIP would correlate to those most resistant to salt (or RNAi, Zn, etc.). There are alternative explanations for a strong X-ChIP signal (e.g. a very high percentage of cells in the population have that particular site bound, even if weakly so). I think more should be made of this point as it is in my view a larger advantage to the technique than what is currently emphasized.

Response: We ranked the 51,000 CTCF motifs in old Figure 3C by X-ChIP signal, and grouped them into three tertiles, representing CTCF motifs with high, medium, or low X-ChIP signal, as shown below.

Using the same analysis in Figure 3H, we compared the performance of S-score and X-ChIP signal in explaining the presence of U-motif within CTCF motifs, which correlates with high CTCF retention on native chromatin. As shown below, in each tertile, the metric of X-ChIP signal exhibits very weak correlation with U-motif presence (compared with dashed diagonal line representing no correlation), while the rank of S-score well explains the presence of U-motif.

Likewise, we further evaluated the performance of S-score and X-ChIP signal in explaining the TPA resistance of CTCF motifs (shown below). And again, S-score performs better than X-ChIP signal in reflecting CTCF motifs' refractory to zinc depletion, in all tertiles.

Thus, we concluded that although the X-ChIP signal and S-score of CTCF motifs exhibit seemingly high correlation globally, the metric of S-score supercedes X-ChIP signal in a fine scale, in reflecting CTCF retention on native chromatin. The above analysis has been added to the revised manuscript (new Figure 3C), and we thank the reviewer for the very helpful discussion.

2. Could the authors comment on the rationale for the different proteinase K used in loMNase-seq versus sequential N-ChIP?

Response: In loMNase-seq, the amount of proteinase K is meant for digestion of proteins in the whole nuclear lysate. In the cases of N-ChIP and sequential N-ChIP, we tuned down the amount of proteinase K based on the fact that the protein level in the pull-down fraction is orders of magnitude lower than the nuclear fraction in loMNase-seq. Nevertheless, the amounts were empirically determined and we hope the reviewer accept the rationale we present here.

3. In Figure 2A, the gained peaks, particularly for ZNF143, are very interesting. I would be very surprised if epitope masking explains such a large number of peaks for ZNF143. The presence of gained peaks at all does not align with the author's framework of utilizing native techniques to avoid 'over-calling' of peaks by using formaldehyde. It would be very interesting to look further into the characteristics of these sites. I am also curious if CTCF is an outlier here without a significant number of gained peaks or not, and would like to see the data for more transcription factors than just these three.

Response: We compared the average ChIP signal between the 2,029 N-ChIP unique and the 3,062 shared ZNF143 sites shown in Figure 2A. Many other TFs (BDP1, MEIS1, STAT5A, ...) exhibit much higher occupancy level on the unique peak set than the shared set (shown below), in agreement with an "epitope masking" hypothesis that too many co-binding proteins, under crosslinking conditions, will potentially hinder ZNF143 antibody's access to the epitope. Interestingly, the occupancy level of CTCF and Cohesin on the N-ChIP unique peaks is much higher than the shared peaks, seemingly suggesting a functional diversification between the two sets of ZNF143 native peaks (shown below).

For the same analysis of other TFs beyond the three in Figure 2, we used motif-annotated loMNase peaks as surrogate N-ChIP peaks for other TFs, and compared them with their respective X-ChIP peaks obtained from ENCODE. The reviewer seems right that CTCF appears to be the only TF showing a great overlap between native and X-ChIP peaks (shown below). Many other TFs exhibit a large subset of peaks unique to native chromatin (shown below). A possible explanation for these native specific peaks is that the chromatin digestion by MNase may make the epitope on these TFs accessible to their antibodies, which does not occur very often under crosslinking conditions. So again, it might be an accessibility issue.

Due to the somehow speculative nature of the above discussion, we did not add the analysis to the revised manuscript. The readers can have access to the above analysis through the journal's policy of publishing review reports alongside the manuscript. We thank the reviewer for triggering an interesting discussion.

4. The authors introduce a previously published N-ChIP data set in Figure 2C. As the development of a new methodology is a significant result of this paper, I would like to have more discussion about what the current method improves over previous methods performing occupancy assays on native chromatin, like the one introduced here.

Response: This dataset is from [PMID: 28079019], and is the only published CTCF N-ChIP dataset in K562 cells as far as we know. We made two major improvements in our N-ChIP protocol: 1) At the cell lysis step, we used a much milder detergent recipe (0.1% NP-40, 0.1% Tween-20, 0.01% digitonin) than the published protocol (0.05% SDS, 1% Triton X-100). 2) A DNA size selection step was enforced after pull-down in our N-ChIP protocol. These two improvements either possibly preserved a higher fraction of CTCF occupancy events during the lysis, or made our library more enriched with CTCF footprint fragments, giving rise to a much higher efficiency of finding peaks, without sacrificing the high motif content, in our dataset than the published dataset (Figure 2C).

5. Figure 6C should not be a line plot because the x-axis is categorical.

Response: We thank the reviewer for the kind reminder, and have made the correction as shown below. Due to the significant overlap between the dots, we changed the figure format to bar graph.

6. Additional commentary on the mechanism by which SUMOylation of CTCF influences its occupancy would add to the story.

Response: To investigate the relationship between SUMOylation and CTCF retention on native chromatin, we used a small compound ML-792 to inhibit SUMOylation, and observed no obvious changes in CTCF N-ChIP signals (Fig. EV4J), suggesting that SUMOylation may be a consequence instead of a cause for CTCF retention on native chromatin. The data was also shown under Reviewer 2, Comment 9.

7. I have concerns about comparing CTCF binding sites between different cell types (K562 versus LNCaP), although in their defense they did get good overlap of RNAi- and salt-resistant sites. It would be nice to see the heat maps to see if it is the 'most' salt-resistant sites (persist at highest salt concentration) that are also the RNAi-resistant sites.

Response: As shown below, we sorted the 19,575 CTCF motifs from 20,936 K562 N-ChIP peaks in Figure 5A by S-score, and the knocked-down CTCF ChIP signal in LNCaP cells shows a concomitant decrease (left). The cumulative analysis also shows that the spectrum of CTCF motif S-score in K562 cells well explains the RNAi resistance of motifs in LNCaP cells (right).

Reviewer #2:

In this manuscript, the authors optimize a low MNase native method that can enrich for small fragments, and then use this in NChIP to look at native factor binding. The authors focus on CTCF and find CTCF dynamics in native chromatin, showing how increasing salt will remove a subset of CTCF bound sites and observe correlation with SUMOylation levels. Authors also examine TPA treatment (zinc depletion) and observe differential binding and microC profiles. Together, this manuscript has many

interesting advancements, although many findings are largely descriptive.

Response: We thank the reviewer for summarizing our work, and for the kind words on our study.

1) With loMNase-seq, it is hugely surprising to me that there are no nucleosome size reads, as I would have anticipated that even without crosslinking the majority of MNase digested chromatin would result in nucleosomes and not only TF footprints. I do not have a question per se, but I would request that the authors offer an explanation to the lack of nucleosome sized reads.

Response: In line with the purpose of identifying TF footprints, we enforced a DNA size selection step during loMNase-seq library preparation to remove the unwanted nucleosomal fragments, and to only keep the short fragments, mostly under 50 bp. Below we show a typical MNase digestion result in loMNase-seq experiments. Due to the trace amount of small fragments, after size selection, the electrophoresis was usually not practiced, and therefore the gel image was not shown.

2) It is unclear to me in the analysis if the authors do computational size selection for any of the analysis performed?

Response: We only did computational size selection (≤ 120 bp) for the digestion boundary analysis (Figure 2E & EV3H). A statement was added to the Methods part in the revised manuscript.

3) Are experiments spike in normalized? For many results there are global losses or reductions in protein occupancy and therefore it is important to establish spike in

normalization for global changes.

Response: The usage of spike-in controls for ChIP experiments is more suited for global and proportional changes of protein occupancy on chromatin (eg., histone modifier inhibition) than for global but uneven alterations to protein occupancy across the genome [PMID: 24709819, 27875550]. We incorporated spike-in controls during both loMNase-seq and N-ChIP experiments in TPA treatment. As shown below, the application of spike-in normalization in loMNase-seq data does not significantly change the overall degree of reduced occupancy on CTCF motifs under TPA treatment, and only has a mild effect on overall reduction in CTCF N-ChIP.

Furthermore, un-normalized N-ChIP signal exhibits good reproducibility between replicates, and shows a comparable degree of reduction after TPA treatment (shown below, upper). Whereas, the spike-in normalized N-ChIP signal exhibits great variation between replicates (shown below, lower), presenting a technical dilemma to pool the data from two sets of replicates.

Lastly, the reduction of occupancy does not appear to be at the same ratio across individual CTCF motifs. As we show below, subsets of CTCF motifs with similar loMNase-seq or N-ChIP signal but with a wide spectrum of S-score, can have very distinct degree of occupancy reduction under TPA treatment.

After weighing all the above factors, we concluded that in TPA treatment, the spike-in normalization does not offer obvious benefits over the potential extra biases it may introduce. Thus, we decided not to use the spike-in normalization for both loMNase and N-ChIP data analysis under TPA treatment.

4) The authors present a lot of data but the reproducibility is not clear to me. How many replicates for experiments and how robust are the results (present PCA plots or even better XY scatterplots for replicates).

Response: We thank the reviewer for pointing out the insufficiency of data reproducibility analysis in our manuscript. In addition to the reproducibility analysis we already had for N-ChIP and sequential N-ChIP (Figure EV2B, Figure 3A, Figure EV3A, Figure EV4H), we added more analysis for the loMNase-seq, N-ChIP and Micro-C datasets from TPA treatment in the revised manuscript (Figure EV6A, B, F, G, EV7A, B, and EV8A). As shown below, for both loMNase-seq and N-ChIP datasets, under DMSO or TPA treatment, the two biological replicates are always highly correlated (right), and exhibit subtle variation along the most contributing PC1 axis in PCA analysis (left), and the two experimental conditions are well segregated in both analyses.

For Micro-C datasets, the high reproducibility between biological replicates was confirmed by PCA analysis (shown below, upper), and by three highly cited methods, at different binning sizes (shown below, lower).

5) In Figure 1, the authors present global motifs from loMNase-seq. Can the authors get more from these data, for example perform peak calling for only promoter locations? Or look over other ChIPseq datasets (for any factors of interest) for intensity of reads?

Response: We thank the reviewer for the helpful suggestion. In the original Figure EV1D, we show 20%-30% overlap of loMNase-seq peaks with ChIP-seq peak sets of 30 different TFs. As shown below and in the revised manuscript, the loMNase-seq peaks widely distributes across the 11 ChromHMM chromatin categories [PMID: 21441907] (shown below), and the loMNase-seq signal significantly enriches at many TF ChIP-seq peak sets (Figure EV1E).

6) On lines 113-115 the two statements seem contradictory to me.

Response: We rephrased the sentence in the revised manuscript, and hope the reviewer find the statement accurate.

7) For data presented in Fig 2 and beyond for NChIP experiments, I do not see any negative controls (no antibody or IgG, for example). Were these included to show background and to subtract for peak calling etc?

Response: We developed N-ChIP by incorporating antibody pull-down steps into loMNase-seq procedures, achieving the capture of footprints left by TF of interest out of a whole set of footprints on native chromatin. That said, this collection of footprints after loMNase-seq digestion but before pull-down steps serves as an "input" for our N-ChIP method. We prepared an input library and indeed, our N-ChIP input looks like loMNase-seq, when viewed in a genome browser (shown below).

We went on to call peaks from the input library using the same parameters as we used for N-ChIP or loMNase-seq peaks. The input peaks show significant overlap with peaks from CTCF N-ChIP or loMNase-seq (shown below), suggesting that the arbitrary normalization using input will lead to a great loss of genuine peaks in N-ChIP dataset.

In a pioneering work of native chromatin by Henikoff Lab [PMID: 24336359], they

made similar observations and concluded that "The resulting occupied regions of genomes ... provide high-resolution maps ... that are not biased toward accessible chromatin and that do not require input normalization". Possibly for the same reasons, they did not mention the need for normalization using IgG either. Theoretically, a library from IgG pull-down will inherit most of the enriched regions from the input, although in a much less extent. Thus, we did not use either input or IgG normalization for our N-ChIP datasets. Finally, as we stated in the Methods part, we used a set of blacklisted regions from ENCODE [PMID: 31249361] to remove those anomalous regions in all our analysis.

8) In Figure 3, the authors present progressive loss of CTCF from chromatin in increasing salt conditions. It would be great to show these results using biochemical assays, for example doing chromatin fractionation to examine whether CTCF is lost from chromatin using this orthogonal assay.

Response: We thank the reviewer for the helpful suggestion. We extracted the chromatin fractions using a protocol in [PMID: 22817891], and incubated the chromatin in the same set of different NaCl concentrations (75, 150, 225 mM) as we used for CTCF N-ChIP. The soluble (supernatant) and insoluble (chromatin) fractions were then collected separately to contain CTCF proteins stripped off and remaining on chromatin respectively. As shown below and in Fig. EV3D, higher NaCl concentration stripped more CTCF proteins from chromatin into supernatant, in agreement with the results from our CTCF N-ChIP data.

9) The authors data presented in Fig 4 suggest that SUMOylation on CTCF might

facilitate retention of chromatin, and these is anticorrelated with transcription. It would be of great interest to investigate this hypothesis. For example, could the authors mutate the residues on CTCF to prevent SUMOylation and then test binding?

Response: As far as we know, previous studies have documented very limited residues on CTCF protein to be potentially SUMOylated [PMID: 19029252]. To investigate the relationship between SUMOylation and CTCF retention on native chromatin, we used a small compound ML-792 to inhibit SUMOylation globally, and observed no obvious changes in CTCF N-ChIP signals (Fig. EV4J and shown below), suggesting that SUMOylation may be a consequence instead of a cause for CTCF retention on native chromatin.

10) I think that Figure 5 is a very powerful figure, in terms of the retention and conservation across cell lines.

Response: We thank the reviewer for this very encouraging comment.

Reviewer #3:

In this manuscript, Hu and colleagues use a low-MNase-concentration variant of native (N) ChIP to look at presumed retention times of TFs on chromatin. Overall, the manuscript is well written, the experiments well explained, and the development of the S-score metric is really clever and potentially very useful. However, I have a couple of important conceptual issues with how the manuscript is presented. First, the use of low MNase tigers for performing native ChIP not new -- many labs will titrate their MNase to suit their target, and the usefulness of low MNase treatments for uncovering labile TF binding footprints has been exemplified before (e.g., in work

by the Laengst lab). Second, footprinting aside, the actual N-ChIP data focus on three ZF TFs known for their strong chromatin association. Even more so, the vast majority of work in the paper only analysis CTCF, arguably the most stably-associated TF on chromatin. This puts into question the usefulness of the whole approach for any other TF (like p65, p53, etc.) that are notoriously labile in the chromatin association, not to mention proteins more promiscuous binding nature like HMGBs. Therefore, I would suggest that there are two avenues the authors can take (in my opinion): either provide data from a more "classical" TF to broaden the scope of the approach and show how their S-score performs in this case or thoroughly refocus the existing data to describe all their nice CTCF-related findings and refrain from generalisations that are not backed by data. In either case, following the suggested changes, the manuscript would be a very nice contribution to the field.

Response: We thank Dr. Papantonis for the very nice comments and suggestions. Regarding the experimental conditions, we had a discussion on the difference between our N-ChIP and a published N-ChIP method when addressing Reviewer #1, Comment #4. With respect to the scope of our study, although the second option would be much easier to accomplish than the first one, we took this opportunity to explore the expandability of our method and analytical framework. As the reviewer suggested, we first investigated p65/RELA as an example of labile TFs. Grouping RELA motifs into four groups by ranking loMNase-seq signal did not disclose an obvious difference in RELA motif sequence conservation (shown below), suggesting that a quantitative metric like S-score may not associate with functional diversification of RELA motifs.

Because of the enrichment of zinc finger TF footprints in our loMNase-seq data, and their important roles in regulating genome organization, we performed a similar set of N-ChIP experiments for MAZ, under 50, 100, or 150 mM NaCl concentration. A similar subsetting effect was observed for MAZ, as in the case of CTCF (Figure EV8B and shown below).

An S-score was established for MAZ to reflect its retention on native chromatin. The S-score successfully delineates a spectrum of differential stability of MAZ-mediated chromatin loops, only when MAZ works independent of CTCF (Figure EV8H, I, and shown below), suggesting a mechanism through division of labor across different TFs in regulating genome organization.

This whole set of new analysis on MAZ is presented in the revised manuscript as a new separate figure (Figure EV8). Accordingly, we narrowed down the generalization of our findings to "zinc finger transcription factors" throughout the title and the main text in the revised manuscript.

Secondary remarks:

Intro: the statement of crosslinking equally stabilizing short- and long-lived TF binding events in the course of 10 min is exaggerated and should be rephrased. Shorter-lived events will consistently have a lower probability of being stabilised and therefore will represent lower abundance/signal intensity events in a ChIP experiment.

Response: We thank the reviewer for the kind suggestion, and have toned down the words in the Introduction and Discussion parts of the revised manuscript. We hope the reviewer find them appropriate.

Results: there are a lot of average plots shown but not a single actual browser view provided so as to be able to assess the signal structure in the N-ChIP experiments. The same applies to Micro-C data, which makes it hard to really assess the TPA treatment effects.

Response: We have added a screenshot example in Fig. 7C and as shown below, to exhibit the changes in loMNase-seq and CTCF N-ChIP signals, and in chromatin contact frequency, under TPA treatment.

9th Apr 2024

Manuscript Number: MSB-2023-12136R

Title: A continuum of zinc finger transcription factor retention on native chromatin underlies dynamic genome organization

Dear Dr Xu,

Thank you for the submission of your revised manuscript to Molecular Systems Biology. We have now received the enclosed reports from the referees that were asked to re-assess it. As you will see the reviewers are now globally supportive and I am pleased to inform you that we will be able to accept your manuscript pending the following final amendments:

1) We require an institutional email address in the manuscript and in our submission system for corresponding authors - currently Dr. Chenhuan Xu does not have an institutional email address listed. It is an option to keep an additional/non-institutional email as a secondary email address. This has recently been made mandatory for all EMBO Press journals, and is included as a requirement upon submission (for more information please see the authorship guidelines in our guide to authors: <https://www.embopress.org/page/journal/17574684/authorguide>).

We can also link your ORCID ID in our system to a profile displaying both email addresses.

2) Please format the Data availability section in the main manuscript text according to the example below:

"The datasets and computer code produced in this study are available in the following databases:

- Chip-Seq data: Gene Expression Omnibus GSE46748 (<https://www.ncbi.nlm.nih.gov/geo/query/acc.cgi?acc=GSE46748>)
- Modeling computer scripts: GitHub (<https://github.com/SysBioChalmers/GECKO/releases/tag/v1.0>)
- [data type]: [full name of the resource] [accession number/identifier] [(doi or URL or identifiers.org/DATABASE:ACCESSION)]"

3) Data availability: Please ensure that the sequencing datasets deposited in GSA are made publicly accessible.

4) Code: Please include a README file on Github with practical use instructions for potential future users of your code.

5) Please rename "Conflict of interest" to "Disclosure and competing interests statement". We updated our journal's competing interests policy in January 2022 and request authors to consider both actual and perceived competing interests. Please review the policy <https://www.embopress.org/competing-interests> and update your competing interests if necessary.

6) In the Materials and Methods, please take care of the following:

- Please rename the "Methods" to "Materials and Methods"
- Cell lines: Please include all information requested in the author checklist for cell lines used in the manuscript (accession number in repository or supplier name, catalog number, clone number, and/or RRID). Currently this information seems to be missing from the Materials and Methods.
- Please be sure to include a sentence in the Materials and Methods as to whether or not the cell lines were recently authenticated and tested for mycoplasma contamination. Please also update the Author Checklist to indicate that this was included in the manuscript.
- Please ensure that a statement on whether or not blinding was done is included in the Materials and Methods even if no blinding was done.
- Antibodies: please ensure that company name, catalog number, and dilutions/amounts of each antibody are reported in the Materials and Methods. Currently dilutions/amounts are missing from the western blot section and the ChIP sections.

7) Please place individual sections of the manuscript in the following order: Title page - Abstract & Keywords - Introduction - Results - Discussion - Materials & Methods - Data Availability - Acknowledgements - Disclosure and Competing Interests Statement - References - Figure Legends - Expanded View Figure Legends.

8) For the figures and figure legends, please take care of the following:

- There are currently 8 EV figures - please reduce this to maximum 5. The remaining 3 figures can be compiled in an appendix file, with the legends under each figure, and renamed Appendix Figure S1, S2, and S3. The appendix should be uploaded in PDF format and needs a table of contents with page numbers. Please also be sure to update the callouts for the appendix figures in the main manuscript text.
- Please indicate the statistical test used for data analysis in the legend of figure EV 1c.
- Please note that the box plots need to be defined in terms of minima, maxima, centre, bounds of box and whiskers, and percentile in the legends of figures 1d, 4c, e; 7g; EV 3j-k; EV 4a-b; EV 6e; EV 7f; EV 8d.
- Please note that information related to n is missing in the legends of figures 1d; 4c, e; 7g; EV 2d; EV 3j-k; EV 4a-b; EV 6e; EV 7a, f; EV 8d.

9) Tables: Please rename Table EV1 as Dataset EV1. Please include a legend for this dataset in the excel file in a separate tab. Please also be sure to update the callout for this table in main manuscript text.

10) Please ensure that all funding sources are entered into the manuscript submission system (i.e. please add Ministry of Science and Technology of China (2022YFC2703303 and 2020YFA0803401), and the National Natural Science Foundation of China (32070611 and 32370624))

11) Synopsis:

- Synopsis image: Please provide a synopsis image that summarises the main findings of the manuscript on a glance. Please

upload it as a high-resolution jpeg file 550 pixels wide x (250-400) pixels high.

- Synopsis text: Please provide a short standfirst (maximum of 300 characters, including space), limit the bullet points to max. 5 and upload it as a separate .doc file. Please write the bullet points to summarise the key NEW findings. They should be designed to be complementary to the abstract - i.e. not repeat the same text. We encourage inclusion of key acronyms and quantitative information (maximum of 30 words / bullet point). Please use the passive voice.

12) As part of the EMBO Publications transparent editorial process initiative (see our policy here:

https://www.embopress.org/transparent-process#Review_Process), Molecular Systems Biology will publish online a Peer Review File (PRF) to accompany accepted manuscripts. This file will be published in conjunction with your paper and will include the anonymous referee reports, your point-by-point response and all pertinent correspondence relating to the manuscript. Let us know whether you agree with the publication of the PRF and as here, if you want to remove or not any figures from it prior to publication. Please note that the Authors checklist will be published at the end of the PRF.

13) Please provide a point-by-point letter INCLUDING my comments as well as the reviewer's reports and your detailed responses (as Word file).

I look forward to reading a new revised version of your manuscript as soon as possible.

Yours sincerely,

Poonam Bheda, PhD
Scientific Editor
Molecular Systems Biology

Please click on the link below to submit the revision:

Reviewer #1:

The authors have done well at addressing my comments.

Reviewer #2:

The authors have addressed all my concerns, as well as those raised by the other reviewers. I think this is a strong manuscript that is ready for publication.

Reviewer #3:

I wish to thank the authors for the efforts they invested into revising this manuscript. All of my remarks are now well addressed, and I propose that the work can be published in its current form.

A. Papantonis

9th Apr 2024

Manuscript Number: MSB-2023-12136R

Title: A continuum of zinc finger transcription factor retention on native chromatin underlies dynamic genome organization

Dear Dr Xu,

Thank you for the submission of your revised manuscript to Molecular Systems Biology. We have now received the enclosed reports from the referees that were asked to re-assess it. As you will see the reviewers are now globally supportive and I am pleased to inform you that we will be able to accept your manuscript pending the following final amendments:

1) We require an institutional email address in the manuscript and in our submission system for corresponding authors - currently Dr. Chenhuan Xu does not have an institutional email address listed. It is an option to keep an additional/non-institutional email as a secondary email address. This has recently been made mandatory for all EMBO Press journals, and is included as a requirement upon submission (for more information please see the authorship guidelines in our guide to authors: <https://www.embopress.org/page/journal/17574684/authorguide>).

We can also link your ORCID ID in our system to a profile displaying both email addresses.

I have added my institutional email address as a secondary option to the submission system. The institutional email address is already associated with the current manuscript.

2) Please format the Data availability section in the main manuscript text according to the example below:

"The datasets and computer code produced in this study are available in the following databases:

- Chip-Seq data: Gene Expression Omnibus GSE46748

(<https://www.ncbi.nlm.nih.gov/geo/query/acc.cgi?acc=GSE46748>)

- Modeling computer scripts: GitHub

(<https://github.com/SysBioChalmers/GECKO/releases/tag/v1.0>)

- [data type]: [full name of the resource] [accession number/identifier] ([doi or URL or identifiers.org/DATABASE:ACCESSION])"

We have followed these guidelines and formatted the revised manuscript accordingly.

3) Data availability: Please ensure that the sequencing datasets deposited in GSA are made publicly accessible.

This submission has been made publicly available. Accordingly, the reviewer link was disabled and removed from the revised manuscript.

4) Code: Please include a README file on Github with practical use instructions for potential future users of your code.

A README file has been added to the Github repository.

5) Please rename "Conflict of interest" to "Disclosure and competing interests statement". We updated our journal's competing interests policy in January 2022 and request authors to consider both actual and perceived competing interests. Please review the policy <https://www.embopress.org/competing-interests> and update your competing interests if necessary.

We have updated this section per instructions above.

6) In the Materials and Methods, please take care of the following:

- Please rename the "Methods" to "Materials and Methods"

- Cell lines: Please include all information requested in the author checklist for cell lines used in the manuscript (accession number in repository or supplier name, catalog number, clone number, and/or RRID). Currently this information seems to be missing from the Materials and Methods.

- Please be sure to include a sentence in the Materials and Methods as to whether or not the cell lines were recently authenticated and tested for mycoplasma contamination. Please also update the Author Checklist to indicate that this was

included in the manuscript.

- Please ensure that a statement on whether or not blinding was done is included in the Materials and Methods even if no blinding was done.

- Antibodies: please ensure that company name, catalog number, and dilutions/amounts of each antibody are reported in the Materials and Methods. Currently dilutions/amounts are missing from the western blot section and the CHIP sections.

We have updated the Materials and Methods section and the Author Checklist per instructions above.

7) Please place individual sections of the manuscript in the following order: Title page

- Abstract & Keywords - Introduction - Results - Discussion - Materials & Methods - Data Availability - Acknowledgements - Disclosure and Competing Interests Statement - References - Figure Legends - Expanded View Figure Legends.

We confirm that the revised manuscript follows this order.

8) For the figures and figure legends, please take care of the following:

- There are currently 8 EV figures - please reduce this to maximum 5. The remaining 3 figures can be compiled in an appendix file, with the legends under each figure, and renamed Appendix Figure S1, S2, and S3. The appendix should be uploaded in PDF format and needs a table of contents with page numbers. Please also be sure to update the callouts for the appendix figures in the main manuscript text.

- Please indicate the statistical test used for data analysis in the legend of figure EV 1c.

- Please note that the box plots need to be defined in terms of minima, maxima, centre, bounds of box and whiskers, and percentile in the legends of figures 1d, 4c, e; 7g; EV 3j-k; EV 4a-b; EV 6e; EV 7f; EV 8d.

- Please note that information related to n is missing in the legends of figures 1d; 4c, e; 7g; EV 2d; EV 3j-k; EV 4a-b; EV 6e; EV 7a, f; EV 8d.

Amendments have been done to meet all above requirements.

9) Tables: Please rename Table EV1 as Dataset EV1. Please include a legend for

this dataset in the excel file in a separate tab. Please also be sure to update the callout for this table in main manuscript text.

We have renamed Table EV1 into Dataset EV1 and included a legend in this file.

10) Please ensure that all funding sources are entered into the manuscript submission system (i.e. please add Ministry of Science and Technology of China (2022YFC2703303 and 2020YFA0803401), and the National Natural Science Foundation of China (32070611 and 32370624))

This information has been added to the submission system.

11) Synopsis:

- Synopsis image: Please provide a synopsis image that summarises the main findings of the manuscript on a glance. Please upload it as a high-resolution jpeg file 550 pixels wide x (250-400) pixels high.

- Synopsis text: Please provide a short standfirst (maximum of 300 characters, including space), limit the bullet points to max. 5 and upload it as a separate .doc file. Please write the bullet points to summarise the key NEW findings. They should be designed to be complementary to the abstract - i.e. not repeat the same text. We encourage inclusion of key acronyms and quantitative information (maximum of 30 words / bullet point). Please use the passive voice.

The synopsis image and text were submitted along with the revised manuscript.

12) As part of the EMBO Publications transparent editorial process initiative (see our policy here: https://www.embopress.org/transparent-process#Review_Process), Molecular Systems Biology will publish online a Peer Review File (PRF) to accompany accepted manuscripts. This file will be published in conjunction with your paper and will include the anonymous referee reports, your point-by-point response and all pertinent correspondence relating to the manuscript. Let us know whether you agree with the publication of the PRF and as here, if you want to remove or not any

figures from it prior to publication. Please note that the Authors checklist will be published at the end of the PRF.

We agree with the policy to publish referee reports to enhance the transparency of editorial process.

13) Please provide a point-by-point letter INCLUDING my comments as well as the reviewer's reports and your detailed responses (as Word file).

This point-by-point response was submitted along with the revised manuscript.

I look forward to reading a new revised version of your manuscript as soon as possible.

Yours sincerely,

Poonam Bheda, PhD
Scientific Editor
Molecular Systems Biology

Please click on the link below to submit the revision:

Reviewer #1:

The authors have done well at addressing my comments.

We thank the reviewer for the very helpful comments.

Reviewer #2:

The authors have addressed all my concerns, as well as those raised by the other reviewers. I think this is a strong manuscript that is ready for publication.

We thank the reviewer for the very helpful comments.

Reviewer #3:

I wish to thank the authors for the efforts they invested into revising this manuscript. All of my remarks are now well addressed, and I propose that the work can be published in its current form.

A. Papantonis

We thank Dr. Papantonis for the very helpful comments.

15th Apr 2024

Manuscript number: MSB-2023-12136RR

Title: A continuum of zinc finger transcription factor retention on native chromatin underlies dynamic genome organization

Dear Dr Xu,

Thank you again for sending us your revised manuscript. We are now satisfied with the final modifications made and I am pleased to inform you that your paper has been accepted for publication.

Yours sincerely,

Poonam Bheda, PhD
Scientific Editor
Molecular Systems Biology
